# UNDERSTANDING THE ROBUSTNESS OF MULTI-MODAL CONTRASTIVE LEARNING TO DISTRIBUTION SHIFT

**Yihao Xue, Siddharth Joshi, Dang Nguyen, Baharan Mirzasoleiman**
Department of Computer Science,
University of California, Los Angeles
`yihaoxue@g.ucla.edu, sjoshi804@cs.ucla.edu,`
`nguyentuanhaidang@gmail.com, baharan@cs.ucla.edu`

## ABSTRACT

Recently, multimodal contrastive learning (MMCL) approaches, such as CLIP (Radford et al., 2021), have achieved a remarkable success in learning representations that are robust against distribution shift and generalize to new domains. Despite the empirical success, the mechanism behind learning such generalizable representations is not understood. In this work, we rigorously analyze this problem and uncover two mechanisms behind MMCL's robustness: *intra-class contrasting*, which allows the model to learn features with a high variance, and *inter-class feature sharing*, where annotated details in one class help learning other classes better. Both mechanisms prevent spurious features that are over-represented in the training data to overshadow the generalizable core features. This yields superior zero-shot classification accuracy under distribution shift. Furthermore, we theoretically demonstrate the benefits of using rich captions on robustness and explore the effect of annotating different types of details in the captions. We validate our theoretical findings through experiments, including a well-designed synthetic experiment and an experiment involving training CLIP models on MSCOCO (Lin et al., 2014)/Conceptual Captions (Sharma et al., 2018) and evaluating them on shifted ImageNets.

## 1 INTRODUCTION

Learning classifiers that generalize under distribution shifts and across various domains has long been a challenge in machine learning. A key reason is that modern models are highly susceptible to learning simple, domain-dependent spurious features in the training data instead of more complex but generalizable features (Zhu et al., 2016; Sagawa et al., 2019; Xiao et al., 2020; Taori et al., 2020). Recently, Multimodal Contrastive Learning (MMCL) has demonstrated significant robustness in zero-shot image classification. It trains vision and language encoders using a contrastive loss to align representations of paired images and text, while pushing apart the representations of images and texts from different pairs (e.g., CLIP (Radford et al., 2021), ALIGN (Jia et al., 2021)).

Radford et al. (2021) have shown that models trained with CLIP, an exemplar of MMCL algorithms, exhibit better Out-of-Distribution (OOD) generalization (*c.f.* their Fig 13). Specifically, CLIP-trained zero-shot classifiers achieve higher OOD accuracy compared to classifiers with equivalent In-Distribution (ID) accuracy that are trained using various supervised learning techniques, including existing robust methods. Interestingly, the advantage of CLIP diminishes if any label supervision is introduced, e.g., through linear probing with labeled data on CLIP's image encoder (*c.f.* Fig 14 of (Radford et al., 2021)). Both findings suggest that the zero-shot classifier produced by CLIP is more robust to distribution shifts than any classifier trained with label supervision.

However, a comprehensive understanding of the reasons behind does not yet exist in the literature. Existing theoretical studies have only examined MMCL's in-distribution generalization (Nakada et al., 2023; Zhang et al., 2023a), but have not explored its OOD-robustness. In this paper, we aim to explain how CLIP, and more broadly MMCL, produces a zero-shot classifier with superior robustness, while demystifying the contributions of MMCL loss and image captions. This study is conducted by comparing the zero-shot classifier learned through MMCL with classifiers learned via supervised learning. The latter is arguably representative of all other non-zero-shot methods of classifier training, as supervised learning is essentially involved (e.g., end-to-end supervised learning, and linear probing on representations learned by unsupervised/self-supervised algorithms).

Specifically, we demonstrate that the MMCL loss accompanied by rich image captions enables at least two mechanisms providing robustness to zero-shot classification. (1) the *intra-class contrasting* between image-text pairs within the same latent class enables MMCL to easily learn generalizable features with a high variance, when they are annotated by text. In contrast, SL is highly prone to learning simple spurious features instead of the more generalizable features with a higher variance (Sagawa et al., 2020). For example if the majority of cow images with diverse shapes and colors appear on a simple green grass background, SL learns the grass to predict 'cow', but MMCL successfully learns the cow; (2) the *inter-class feature sharing* enabled by MMCL loss allows learning information about a class that only exists and is annotated in other classes. For example, if all the images of the tree class have green leaves but an image in the wolf class has a tree without leaves in its background, MMCL can disassociate the green leaf from the tree class. In contrast, SL cannot leverage this information and learns the green leaves as an indistinguishable part of 'tree'. Hence, it fails to classify trees without green leaves in the test data. Both mechanisms together enable MMCL to learn representations that are robust against distribution shift and generalize to unseen domains.

Furthermore, to emphasize the importance of captions, we analyze the effect of varying caption richness and show that rich captions are essential for achieving robustness. As an extreme case, if the captions are merely labels, no gains in robustness can be achieved.

We further support our theoretical findings with experiments, including a well-designed synthetic experiment and experiments on real datasets, including MSCOCO, Conceptual Captions, and shifted versions of ImageNet. The results demonstrate the crucial roles of the MMCL loss function and rich captions in achieving robustness, further validating our theoretical findings.

## 2 RELATED WORKS

**Distribution shift.** There is a long line of work on dealing with different types of distribution shift. This includes sub-population shift and domain generalization among others, where distribution of sub-populations in training and test data is different, and some sub-populations may be underrepresented or missing in the training data (Cai et al., 2021; Yang et al., 2023; Santurkar et al., 2020), (Gulrajani & Lopez-Paz, 2020; Joshi et al., 2023; Zhang et al., 2023b; Hu et al., 2020; Fahrbach et al., 2023), or a hybrid of both Koh et al. (2021). Another line of research focuses on evaluating models on natural variations in the source of data collection, with the precise category of shift typically unformalized or unknown. For example, a dataset that contains art and cartoon renditions of ImageNet classes (Hendrycks et al., 2021a), and other variations of ImageNet (Barbu et al., 2019; Recht et al., 2019; Shankar et al., 2021). Despite the diversity of settings, extensive studies (Sagawa et al., 2019; 2020; Xiao et al., 2020; Ilyas et al., 2019) revealed a common theme across these subfields: deep learning models often rely heavily on *spurious correlations* that are specific to the training data but do not hold in general, e.g., those between certain object classes and backgrounds/textures in the image (Zhu et al., 2016; Geirhos et al., 2018).

**Multi-modal (contrastive) learning.** Learning better representations based on multiple modalities has been a long pursuit (Ngiam et al., 2011; Srivastava & Salakhutdinov, 2012). Numerous methods for learning joint vision-language representations (Li et al., 2019; Lu et al., 2019; Tan & Bansal, 2019; Li et al., 2020; Yao et al., 2021) have emerged. Among them, MMCL (Radford et al., 2021; Jia et al., 2021; Mu et al., 2022; Goel et al., 2022; Pham et al., 2023) has stood out by achieving SOTA performance in various tasks. Notably, (Radford et al., 2021) showed that MMCL on large image-text datasets achieves a significant improvement in robustness to distribution shift. The empirical investigations of (Fang et al., 2022) suggests that this is only attributed to the large diverse image training data, with MMCL loss and text supervision contributing little. We show *provably* that it is not *only* the diverse image data that contributes to superior robustness of MMCL. Indeed, MMCL loss and richness of text annotations are crucial factors.

## 3 A FRAMEWORK FOR COMPARING MMCL AND SL

In this section, we present a general framework for comparing MMCL and SL, along with the corresponding notations. We start by modeling the multimodal data, and then formalize the MMCL and SL pipelines and their evaluation for robustness to distribution shift. We will formulate and analyze specific types of distribution shift in the next section.

## 3.1 MODELING MULTIMODAL DATA

To model multimodal data, it is essential to capture the fact that inputs from different modalities can represent the same abstract notion. For instance, both text and an image can represent 'a cow on grass'. We define *underlying feature vectors* to model this abstract notion, and model each input in a specific modality as a projection of the underlying feature vector onto that modality's input space.

**Underlying feature .** There is an underlying feature space shared among different modalities, where abstract notions reside. We model this as a vector space $\mathbb{R}^l$, where each vector is termed an *underlying feature vector* (e.g., 'a cow on grass'), and each element within the vector is referred to as an *underlying feature* (e.g., 'cow').

**Latent classes and labels.** Each $z$ is associated with a *latent class*, represented by a *label* $y$. We note that the labels are only used by SL but not by MMCL.

**Inputs in each modality.** Each input example in a modality is an instantiation of an abstract notion. We model this as a projection from an underlying feature vector to another space where this modality's inputs live. Formally, let $M$ represent a modality. Given a underlying feature vector $z$, a corresponding input $x_M$ in this modality is generated as: $x_M = D_M \mu_M(z) + \xi_M$, where $\mu_M(z) \in \mathbb{R}^l$ is a random vector that depends on $z$. It can be interpreted as a possibly distorted version of the original feature vector $z$. Note that setting $\mu_M(z) = z$ implies no distortion in the features when represented in this modality. $\xi_M \in \mathbb{R}^{d_M}$ is a random noise drawn from $\mathcal{N}(0, \frac{\sigma_\xi^2}{d_M} \mathbf{I}_{d_M})$. The matrix $D_M \in \mathbb{R}^{d_M \times l}$ ($d_M > l$) is a matrix with orthonormal columns that can be interpreted as a dictionary matrix. It captures the transformation from the lower dimensional feature space to the higher dimensional input space. Different modalities can have different $D_M$ matrices, reflecting the idea that the same underlying feature may be instantiated differently in each modality (e.g., colors are represented differently in images and texts). Modeling modalities as above is consistent with (Nakada et al., 2023).

In this paper, for clarity and illustration, we focus on the popular vision and language modalities. We let $I$ denote the modality for images, and $T$ denote the modality for texts. However, we note that our framework and results directly apply to other modalities.

**Distribution shift.** We define two joint distributions between underlying features and latent classes: $\mathscr{P}^\star$, representing the 'ground-truth' in the real world, and $\mathscr{P}^{Tr}$, from which our training data are drawn. We let $\mathscr{P}^{Tr}$ exhibit spurious correlations between certain features and latent classes which do not hold in the ground-truth $\mathscr{P}^\star$. This setup captures the underlying reason for the performance drop observed in various types of distribution shift scenarios, as we will discuss in Section 2.

## 3.2 MULTI-MODAL CONTRASTIVE LEARNING (MMCL)

Unlike traditional supervised learning, MMCL does not see the input data's latent classes, but is instead given pairs of inputs from two modalities and aims to learn the correspondence between them.

**Training dataset.** The training dataset comprises $n$ image-text pairs, denoted as $\{(x_{I,i}, x_{T,i})\}_{i=1}^n$, where for each index $i$, both $x_{I,i}$ and $x_{T,i}$ are generated based on the same underlying feature vector $z_i$. In practice, the texts are usually captions accompanying the images. The feature vectors $\{z_i\}_{i=1}^n$ are drawn from the training distribution $\mathscr{C}^{Tr}$.

**Linear encoders.** The encoders for modalities $I$ and $T$ are denoted as $g_I : \mathbb{R}^{d_I} \to \mathbb{R}^p$ and $g_T : \mathbb{R}^{d_T} \to \mathbb{R}^p$ respectively. We consider linear models for the encoders, given by $g_I(x) = W_I x$ and $g_T(x) = W_T x$, where $W_I \in \mathbb{R}^{d_I \times p}$ and $W_T \in \mathbb{R}^{d_T \times p}$ with $p \geq l$ are the corresponding encoder parameters. Linear encoders are employed widely in previous studies of MMCL (Nakada et al., 2023; Ren & Li, 2023) and general feature learning (Jing et al., 2021; Tian et al., 2021; Ji et al., 2021; Wu et al., 2022; Tian, 2022; Xue et al., 2023), to facilitate the analysis. In Section 6, we will empirically confirm that our findings extend to non-linear models.

**Representation learning with MMCL.** MMCL learns representations for both modalities in a shared latent space. We consider the linearized contrastive loss function from Nakada et al. (2023):

$$\mathcal{L}_{\text{MMCL}}(W_I, W_T) = \frac{1}{2n(n-1)} \sum_i \sum_{j \neq i} (s_{ij} - s_{ii}) + \frac{1}{2n(n-1)} \sum_i \sum_{j \neq i} (s_{ji} - s_{ii}) + \frac{\rho}{2} \|W_I^\top W_T\|_F^2,$$

where $s_{ij} := g_I(x_{I,i})^\top g_T(x_{T,j}) = (W_I x_{I,i})^\top W_T x_{T,j}$ is the similarity (measured by inner product) between representations of an image and a text. This loss encourages the model to *align* each

image-text pair by increasing their representation similarity ($s_{ii}$) and *contrast* between images and texts that are not paired together by reducing their representation similarity ($s_{ij}, i \neq j$). The last term is a regularization term with $\rho > 0$. The linear loss and its uni-modal counterpart are widely used in analysis of CL, as they closely captures the dynamics of popular contrastive losses (Ji et al., 2021; Tian, 2022; Nakada et al., 2023), such as CLIP, as we will experimentally confirm in Section 6.

**Prompts for zero-shot classification.** We test the model's capability in zero-shot classification, where a text prompt is created for each label (e.g., 'a photo of a *dog*'), and the prediction is determined by the prompt with the highest representation similarity with the given image. To formalize this, we define the prompt $\boldsymbol{p}_y$ for each latent class $y$ as $\boldsymbol{p}_y = \boldsymbol{D}_T \bar{\boldsymbol{z}}_y$, where $\bar{\boldsymbol{z}}_y := \mathbb{E}_{(\boldsymbol{z},y) \sim \mathscr{P}^\star}[\boldsymbol{z}|y]$. That is, the prompt is 'the center of all underlying feature vectors with label $y$ in the true distribution' represented in modality $T$. This closely resembles real world practices where the representation of multiple texts with engineered templates like 'a bad photo of a {}', 'a good photo of a {}' are averaged (Radford et al., 2021).

**Robustness evaluation.** Given two encoders $g_I$ and $g_T$ with parameters $\boldsymbol{W}_I$ and $\boldsymbol{W}_T$, respectively, we evaluate the zero-shot performance on the true distribution $\mathscr{P}^\star$. Given an image $\boldsymbol{x}_I$, the prediction is $\hat{y}(\boldsymbol{x}_I) = \arg\max_y g_I(\boldsymbol{x}_I)^\top g_T(\boldsymbol{p}_y)$. The test accuracy, denoted by $\text{Acc}_{\mathscr{P}^\star}^{\text{MMCL}}(\boldsymbol{W}_I, \boldsymbol{W}_T)$, is

$$\text{Acc}_{\mathscr{P}^\star}^{\text{MMCL}}(\boldsymbol{W}_I, \boldsymbol{W}_T) = \mathbb{E}_{(\boldsymbol{z},y) \in \mathscr{P}^\star, \boldsymbol{x}_I = \boldsymbol{D}_I \boldsymbol{\mu}(\boldsymbol{z}) + \boldsymbol{\xi}_I}[\mathbb{1}(\hat{y}(\boldsymbol{x}_I) = y)], \tag{1}$$

where $\mathbb{1}(\cdot)$ denotes the indicator function.

**Relation to feature cross-covariance.** We utilize the connection between the cross-covariance between images and captions, and the MMCL objective for our analysis.

**Definition 3.1.** *We define $\boldsymbol{C}^{Tr}$ as the cross-covariance between images' and texts' feature vectors $\boldsymbol{C}^{Tr} := \mathbb{E}_{\boldsymbol{z} \in \mathscr{C}^{Tr}, \boldsymbol{\mu}_I(\boldsymbol{z}), \boldsymbol{\mu}_T(\boldsymbol{z})}[\boldsymbol{\mu}_I(\boldsymbol{z})\boldsymbol{\mu}_T(\boldsymbol{z})^\top]$. When $\boldsymbol{\mu}_I(\cdot)$ and $\boldsymbol{\mu}_T(\cdot)$ both are identity, $\boldsymbol{C}^{Tr}$ is the covariance of the original feature vector.*

**Lemma 3.2** (Informal)**.** *Given an image with feature $\boldsymbol{\mu}'$ and a text with feature $\boldsymbol{\mu}''$, the similarity (inner product of representations) between them, computed using encoders trained on the training set, is: similarity score $\approx \boldsymbol{\mu}'^\top \boldsymbol{C}^{Tr} \boldsymbol{\mu}'' = \sum_{i=1}^l \sum_{j=1}^l C_{ij}^{Tr} \mu_i' \mu_j''$.*

That is, the image-text similarity is a weighted sum of products between the features in image and text inputs. The weights are determined by the feature cross-covariance matrix of training data, whose $i, j$-th element is the covariance between feature $i$ in images and feature $j$ in texts.

**Importance of zero-shot.** We emphasize that using zero-shot classification instead of training a linear classifier on the representations is crucial for achieving robustness in MMCL. The latter essentially involves SL, which falls short for the same reasons as shown in our analysis for SL in Section 4.

### 3.3 SUPERVISED LEARNING (SL)

Standard SL has access to each input's label and the inputs are from a single modality (i.e., images). Let $\{(\boldsymbol{x}_{I,i}, y_i)\}_{i=1}^n$ be the training dataset with $n$ inputs $\boldsymbol{x}_{I,i}$ and their labels $y_i$, we train a linear model $f(\boldsymbol{x}) = \boldsymbol{W}^\top \boldsymbol{x}$ with weights $\boldsymbol{W} \in \mathbb{R}^{d_I \times q}$, where $q = 1$ for binary classification and $q = \#$classes for multi-class classification. We consider minimizing logistic loss for binary classification, and Cross-Entropy loss for multiclass classification, with gradient descent at a sufficiently small step size.

**Robustness evaluation.** Given a model with weights $\boldsymbol{W}$, the accuracy, denoted by $\text{Acc}_{\mathscr{P}^\star}^{\text{SL}}(\boldsymbol{W})$, is evaluated on the true distribiution $\mathscr{P}^\star$ as $\text{Acc}_{\mathscr{P}^\star}^{\text{SL}}(\boldsymbol{W}) = \mathbb{E}_{(\boldsymbol{z},y) \in \mathscr{P}^\star, \boldsymbol{x}_I = \boldsymbol{D}_I \boldsymbol{\mu}(\boldsymbol{z}) + \boldsymbol{\xi}_I}[\mathbb{1}(\hat{y}(\boldsymbol{x}_I) = y)]$, where $\hat{y}(\boldsymbol{x}_I) = \text{sign}(\boldsymbol{W}^\top \boldsymbol{x}_{I_j})$ for binary classification, and $\hat{y}(\boldsymbol{x}_I) = \arg\max_j[\boldsymbol{W}^\top \boldsymbol{x}_I]_j$ for multi-class classification, with $[\boldsymbol{W}^\top \boldsymbol{x}_I]_j$ denoting the $j$-th element in the vector $\boldsymbol{W}^\top \boldsymbol{x}_I$.

## 4 TWO MECHANISMS BEHIND THE ROBUSTNESS OF MMCL

Next, we explore two scenarios illustrating MMCL's superior robustness to distribution shift, compared to SL. First, we consider the scenario where generalizable core feature has a higher variance than domain-dependent spurious feature. Then, we consider the data distribution where each latent class has a core feature, that co-occurs with a strong spurious feature in the training data. These features can occur in other latent classes as well, independently of each other. For clarity, we set $\boldsymbol{\mu}_I(\boldsymbol{z}) = \boldsymbol{z}, \boldsymbol{\mu}_T(\boldsymbol{z}) = \boldsymbol{z}, \forall \boldsymbol{z} \in \mathbb{R}^l$ in this section.

## 4.1 ROBUSTNESS VIA INTRA-CLASS CONTRASTING

We start by analyzing the first scenario which illustrates how MMCL can learn features that are challenging for SL to learn. Consider the case where the majority or all images of a 'cow' appear on 'grass'. Here, grass is a spurious feature with high correlation with cow. Grass is often a simple green surface without a high variation. But, cows can vary a lot in their appearance. This makes cows more difficult to learn than grass. Below, we will formalize this scenario and demonstrate that SL learns the spurious feature (grass) but MMCL learns the generalizable feature (cow) and obtains a superior robustness.

### 4.1.1 DISTRIBUTION OF FEATURES

The following definition simulates the aforementioned scenario.

**Definition 4.1 (Data Model** 1**).** *In both $\mathscr{P}^\star$ and $\mathscr{P}^{Tr}$, each label $y$ is uniformly drawn from $\{-1, 1\}$ and the corresponding feature vector $\boldsymbol{z} \in \mathbb{R}^2$ is generated as $\boldsymbol{z} = [z_{core}, z_{spu}]^T$ where $z_{core} \sim \mathcal{N}(y, \sigma_{core}^2)$, represents the core feature that contains information of the label $y$, and $z_{spu} \sim \mathcal{N}(a, \sigma_{spu}^2)$. In the true distribution $\mathscr{P}^\star$, $a$ is uniformly drawn from $\{-1, 1\}$ and is independent of the label $y$, making the feature $z_{spu}$ irrelevant to the label. However, in the training distribution $\mathscr{C}^{Tr}$, there is a strong correlation between $a$ and $y$, s.t. $\Pr(a = y) = p_{spu}$, where $1 \geq p_{spu} > 1/2$.*

Recall from Section 3.1 that the inputs in each modality are generated based on feature vectors. In SL, where we have only one modality, the situation becomes equivalent to the one analyzed in (Sagawa et al., 2020). Similar variants are studied in (Wald et al., 2021; Aubin et al., 2021; Yao et al., 2022) to investigate distribution shift and out-of-domain generalization. Despite its simplicity, this setup reflects key aspects of general distribution shift. Here, $z_{\text{core}}$ is the core feature and $z_{\text{spu}}$ is the spurious feature, such as 'grass' in the aforementioned example, or texture/backgrounds in ImageNet.

We assume that the core feature has a larger variance than the spurious feature, indicated by the values of $\sigma_{\text{core}}$ and $\sigma_{\text{spu}}$. This is detailed in below, along with some additional assumptions.

**Assumption 4.2.** *The gap between the variances of the core and spurious features is significant: $\sigma_{core} = \Theta(1)$, $\sigma_{core} \geq 1$ and $\sigma_{spu} = O(\frac{1}{\sqrt{\log n}})$. The spurious correlation is large: $p_{spu} = 1 - o(1)$. We consider the high-dimensional (overparameterized) setting where $n = \omega(1)$, $d_I = \Omega(n)$ and $d_T = \Omega(n)$. The noise levels are not too large: $\sigma_{\xi,I} = O(\log n)$ and $\sigma_{\xi,T} = O(\log n)$.*

### 4.1.2 COMPARING ROBUSTNESS OF SL AND MMCL

Under Assumption 4.2, SL tends to associate labels mostly with the spurious feature, as they appear to be more stable and reliable for prediction compared to the core feature. This results in low accuracy when tested on the ground-truth distribution, as demonstrated in the following theorem.

**Theorem 4.3** (Theorem 1 from (Sagawa et al., 2020))**.** *Let $\boldsymbol{W}^*$ represent the model trained using SL as described in Section 3.3. Assuming that Assumption 4.2 holds, and $n$ and $d_I$ are sufficiently large, with a high probability, the accuracy of $\boldsymbol{W}^*$ on the true distribution satisfies $Acc_{\mathscr{P}^\star}^{SL}(\boldsymbol{W}^*) \leq 2/3$. Additionally, the model's test accuracy on examples where $a \neq y$ is $\leq 1/3$, worse than random chance.*

Next, we examine MMCL. From Lemma 3.2, we know that the similarity between an image with feature $\boldsymbol{z}$ and a text with feature $\boldsymbol{z}'$ is approximately $\begin{bmatrix} z_{\text{core}} & z_{\text{spu}} \end{bmatrix} \begin{bmatrix} 1 + \sigma_{\text{core}}^2 & 2p_{\text{spu}} - 1 \\ 2p_{\text{spu}} - 1 & 1 + \sigma_{\text{spu}}^2 \end{bmatrix} \begin{bmatrix} z'_{\text{core}} \\ z'_{\text{spu}} \end{bmatrix}$, showing that the variance of features ensures that image and text features, that share the underlying core features, have a higher similarity score. Furthermore, if we let $\boldsymbol{z}'$ be the feature $\bar{\boldsymbol{z}}_{y'} = [y' \ 0]^\top$ in label $y'$'s corresponding prompt $\boldsymbol{p}_{y'}$, we deduce that the similarity to the prompt is approximately $(1 + \sigma_{\text{core}}^2)y'z_{\text{core}} + (2p_{\text{spu}} - 1)y'z_{\text{spu}}$. Here, the core feature carries more weight when the variance is large. In essence, since the MMCL loss contrasts images and unpaired texts in the same latent classes, learning features that have high variance is encouraged; this is in contrast with SL, where features that have low variance are preferred. With the above observation, after bounding the effect of noise, we arrive at the following theorem (with proof in Appendix C.2).

**Theorem 4.4.** *Let $\boldsymbol{W}_I^*$ and $\boldsymbol{W}_T^*$ be the weights of the encoders trained using MMCL as described in Section 3.2. Under Assumption 4.2 [1], with a high probability of at least $1 - O(\frac{1}{\text{poly}(n)}) = 1 - o(1)$,*

---

[1]The theorem holds under more relaxed assumptions about the variances and spurious correlation level; see details in Appendix C.2), but here we use Assumption 4.2 to keep consistency with Theorem 4.3

*the encoders achieve the following zero-shot accuracy on the true distribution*

$$Acc_{\mathscr{P}^\star}^{MMCL}(\boldsymbol{W}_I^*, \boldsymbol{W}_T^*) \geq 1 - \frac{1}{2}\boldsymbol{\Phi}(\kappa_1) - \frac{1}{2}\boldsymbol{\Phi}(\kappa_2) - o(1),$$

*where* $\kappa_1 = \frac{2p_{spu}-2-\sigma_{core}^2}{\sqrt{(1+\sigma_{core}^2)^2\sigma_{core}^2+(2p_{spu}-1)^2\sigma_{spu}^2}}$, $\kappa_2 = \frac{-2p_{spu}-\sigma_{core}^2}{\sqrt{(1+\sigma_{core}^2)^2\sigma_{core}^2+(2p_{spu}-1)^2\sigma_{spu}^2}}$ *and* $\boldsymbol{\Phi}$ *denotes the CDF of the standard normal distribution. Meanwhile, the model's test accuracy on examples where* $a \neq y$ *is lower bounded by* $1 - \boldsymbol{\Phi}(\kappa_1) - o(1)$.

**Corollary 4.5.** *With* $\sigma_{core} = 1$, *for sufficiently large n,* $Acc_{\mathscr{P}^\star}^{MMCL}(\boldsymbol{W}_I^*, \boldsymbol{W}_T^*) \geq 81\%$, *Moreover, in this case, no model can achieve an accuracy higher than* $85\%$.

This, compared with Theorem 4.3, demonstrates that MMCL can outperform SL by a large margin, and comes close to achieving the best possible accuracy of $85\%$.

Additionally, in terms of performance on examples where the spurious correlation does not hold, i.e., $a \neq y$, it's evident that MMCL excels. As Theorem 4.3 shows, SL's accuracy is even worse than random chance. In contrast, Theorem 4.4 demonstrates that MMCL consistently performs better than random chance. It maintains random chance even in the worst-case scenario, as indicated by $\boldsymbol{\Phi}(\kappa_1) \leq \frac{1}{2}$, owing to $2p_{\text{spu}} - 2 - \sigma_{\text{core}}^2 \leq 0$. When $\sigma_{\text{core}} = 1$, it achieves an accuracy of 69%.

### 4.2 ROBUSTNESS VIA INTER-CLASS FEATURE SHARING

Next, consider the second scenario and demonstrate how MMCL benefits from annotated details in some latent classes to disassociate spurious correlations in other latent classes, while SL fails to grasp these details. For example, typical images of a 'tree' have green leaves. However, trees in in the background of images of 'wolf' or 'ski resort' may appear without leaves. SL, which only observes the labels, tends to overlook the trees without leaves as they do not contribute to learning 'wolf' and 'ski resort', thus incorrectly correlating trees with the color green. In contrast, in MMCL, if the trees without leaves are annotated in the captions, the model can disassociate the green leaves from tree.

#### 4.2.1 DISTRIBUTION OF FEATURES

We first present the underlying feature distributions and then compare MMCL's robustness with SL.

**Definition 4.6 (Data Model** 2**).** *True distribution* $\mathscr{P}^\star$. *We have* $2m$ *latent classes in total, with labels* $1, \ldots, 2m$. *For each label* $y$, *we define a unique alias* $(k, c)$: $k = \lfloor(y+1)/2\rfloor$, *and* $c = 1$ *if* $y$ *is odd, and* $c = -1$ *if* $y$ *is even. The label is sampled uniformly. Let* $\beta \in [0, 1)$. *Given a label alias* $(k, c)$, *the corresponding feature vector* $\boldsymbol{z} = [z_1, z_2, \ldots, z_{2m}]^\top$ *is generated as:*

$$\forall j \leq m, \quad \text{if } j = k \text{ then } z_j = c \qquad \text{if } j \neq k \text{ then } z_j \sim U(\{-\beta, +\beta\})$$
$$\forall j > m, \quad \text{if } j = k + m \text{ then } z_j \sim U(\{-\alpha, +\alpha\}) \quad \text{if } j \neq k + m \text{ then } z_j \sim U(\{-\beta\alpha, \beta\alpha\})$$

*where* $U(S)$ *denotes the uniform distribution over set* $S$.

*Training distribution* $\mathscr{P}^{Tr}$. *The training distribution is similar to the true distribution, but with* $z_{k+m}$ *always equal to* $c\alpha$, *making it appear as if the* $k + m$-*th coordinate also indicates the label.*

Here, each feature vector in latent class $(k, c)$ (e.g., 'tree') has a core feature at coordinate $k$ (characteristics of a tree) and a spurious feature at coordinate $k + m$ that correlates with the latent class in the training distribution but not the true distribution (e.g., the color green). With a large $\alpha$, such a spurious feature has a larger magnitude than the true feature, making it easier to be learned. There are also other features at different coordinates that do not correlate with the label; these features are weaker (indicated by $\beta < 1$) so that they do not change the latent class. One observation is that examples in latent class other than $(k, c)$ would show no correlation between the $k$-th and $k + m$-th features, hinting at their independence from each other (e.g., trees are not necessarily green). We will show that unlike SL, MMCL can leverage such a hint to obtain a superior robustness.

#### 4.2.2 COMPARING ROBUSTNESS OF SL AND MMCL

The theorem below demonstrates that SL achieves a low accuracy under distribution shift when the spurious feature is strong, i.e., when $\alpha$ is large.

**Theorem 4.7.** *Assuming that the input noise in each modality is zero, i.e., $\sigma_{\xi,I} = \sigma_{\xi,T} = 0$, and all possible feature vectors in $\mathscr{P}^{Tr}$ uniformly appear in the training dataset.* [2] *Let $\boldsymbol{W}^*$ be the model trained using SL as described in Section 3.3. The accuracy on the true distribution has the following upper bound: $Acc_{\mathscr{P}^\star}^{SL}(\boldsymbol{W}^*) \leq 50\% + \frac{2}{(1+\alpha^2)(1-\beta)^2-8}$. For example, if $\alpha = 10$ and $\beta = 1/3$, then $Acc_{\mathscr{P}^\star}^{SL}(\boldsymbol{W}^*) \leq 60\%$.*

Next, we will examine how MMCL leverages the information about independence of core and spurious features in each latent class, which is hidden in other latent classes. First, recall the conclusion in Lemma 3.2, and obtain that the similarity between an image with features $\boldsymbol{z}$ and a text with features $\boldsymbol{z}'$ is given by $\boldsymbol{z}^\top \begin{bmatrix} \frac{1+(m-1)\beta^2}{m}\mathbf{I}_m & \frac{\alpha}{m}\mathbf{I}_m \\ \frac{\alpha}{m}\mathbf{I}_m & \frac{1+(m-1)\beta^2}{m}\alpha^2\mathbf{I}_m \end{bmatrix} \boldsymbol{z}'$. The fact that $\beta$ only appears on the diagonal and not on off-diagonal elements shows that the occurrence of core feature of a given latent class in other latent classes increases the weight for same-feature products between images and texts rather than different-feature products. For example, trees without green leaves in classes other than tree increase the covariance between texts and images of tree, but do not contribute to the correlation between tree and green. Hence appearance of green in any image has a limited impact on its similarity to a text describing a tree. More precisely, when computing the similarity between a given image and the prompt for a tree, a weight $\frac{1+(m-1)\beta^2}{m}$ is assigned to 'the true characteristic of a tree' and a weight $\frac{\alpha}{m}$ is assigned to 'green'. Here, a larger $\beta$ leads to more weight placed on the core feature, highlighting how MMCL utilizes shared features between classes to enhance robustness. This insight leads us to the following theorem demonstrating the superior performance of MMCL under distribution shift.

**Theorem 4.8.** *Under the same assumption as in Theorem 4.7 Let $\boldsymbol{W}_I^*$ and $\boldsymbol{W}_T^*$ be the weights of encoders trained using MMCL as described in Section 3.2. Then as long as $\beta^2 m > \frac{\alpha^2(1+\beta)}{1-\beta} - 1 + \beta^2$, the model has 100% zero-shot accuracy on the true distribution, i.e., $Acc_{\mathscr{P}^\star}^{MMCL}(\boldsymbol{W}_I^*, \boldsymbol{W}_T^*) = 100\%$.*

We also observe that if the features were not shared between classes, i.e., $\beta = 0$, it would be impossible for the model to achieve such performance. This once again emphasizes the role of shared features.

**Important Consideration about Robustness**   An important question is whether the improvement in accuracy under distribution shift is solely due to MMCL's improvement in in-distribution generalization. In Appendix E, we demonstrate that we control for in-distribution generalization in both theoretical examples. Specifically, in Data Model 1, SL has slightly better in-distribution accuracy, while in Data Model 2, both SL and MMCL achieve 100% in-distribution accuracy. Thus, MMCL's improvement solely results from enhanced robustness, and in fact, both relative and effective robustness as defined in Taori et al. (2020).

## 5   UNDERSTANDING THE BENEFIT OF RICH IMAGE CAPTIONS

In Section 4, we assumed that both $\boldsymbol{\mu}_I(\cdot)$ and $\boldsymbol{\mu}_T(\cdot)$ are identity, implying that the captions mentioned everything depicted in the image. However, in practice, captions often serve as annotations or illustrations accompanying the image, with certain details omitted. Empirical evidence suggests that rich captions are generally beneficial (Santurkar et al., 2022; Nguyen et al., 2023), but it remains unclear if richness of captions can affect robustness and, if so, how. In this section, we theoretically investigate this question by varying how much and what information is mentioned in captions. Specifically, we keep $\boldsymbol{\mu}_I(\cdot)$ as an identity function, while let $\boldsymbol{\mu}_T(\boldsymbol{z})$ represent a masked version of the original feature vector $\boldsymbol{z}$, where some information may not be reflected in caption.

**Benefits of mentioning variations in the core features.** Recall that in Section 4.1, utilizing Data Model 1 (Definition 4.1), we showed that MMCL can learn large-variance core features better than SL, resulting in less reliance on the spurious feature. Now, we use the same data model to explore what happens if the feature variance is not fully reflected in the captions. For example, when the caption only contains the word 'cows' or 'grass', without describing their appearance.

**Definition 5.1** (Feature masking in data model 1 (Definition 4.1)). *Given a feature vector $\boldsymbol{z} = \begin{bmatrix} z_{core} \\ z_{spu} \end{bmatrix}$ with corresponding $y$ and $a$, we let $\boldsymbol{\mu}_T(\boldsymbol{z}) = \begin{bmatrix} y + \psi_{core}(z_{core} - y) \\ a + \psi_{spu}(z_{spu} - a) \end{bmatrix}$, with $\psi_{core}$ drawn from Bernoulli($\pi_{core}$) and $\psi_{spu}$ drawn from Bernoulli($\pi_{spu}$). Both $\pi_{core}$ and $\pi_{spu}$ are $\in [0, 1]$.*

---

[2]Assumptions are made to simplify the analysis, but our analysis can be readily extended to show that same conclusions holds with high probability in broader settings with sufficient sample size and reasonable noise level.

Here, $z_{\text{core}} - y$ and $z_{\text{spu}} - a$ represent the variations in the core and spurious features, both of which are Gaussian random variables by Definition 4.1. This implies that the captions capture these variations with probabilities $\pi_{\text{core}}$ and $\pi_{\text{spu}}$, respectively. When $\pi = 0$, caption ignores all the details and treats all features of the same kind as a single entity. The following theorem shows the effects of $\pi_{\text{core}}, \pi_{\text{spu}}$.

**Theorem 5.2.** *With data from data model 1 and $\boldsymbol{\mu}_T$ defined in Definition 5.1, with a high probability, the model trained using MMCL has a test accuracy on examples where the spurious correlation does not hold (i.e., $a \neq y$) given by $1 - \boldsymbol{\Phi}\left(\frac{2p - 2 - \pi_{core}^2 \sigma_{core}^2}{\sqrt{(1 + \pi_{core}^2 \sigma_{core}^2)^2 \sigma_{core}^2 + (2p-1)^2 \sigma_{spu}^2}}\right) \pm o(1)$. The non-negligible part of this accuracy increases as $\pi_{core}$ increases and is independent of $\pi_{spu}$.*

The theorem reveals that the model exhibits less reliance on the spurious correlation when the caption mentions the variance in the core feature (e.g., appearance of the cow in each specific image). Additionally, we notice that mentioning variance in the spurious feature has minimal effect on the robustness, as it does not impact the correlation with the core feature.

**Mentioning more features benefits robustness.** Next, we utilize data model 2 to explore the effect of mentioning more features in the captions.

**Definition 5.3** (Feature masking in data model 2 (Definition 4.6))**.** *For a feature vector $\boldsymbol{z}$ with label $(k, c)$, let $\boldsymbol{\mu}_T(\boldsymbol{z}) = \boldsymbol{\psi} \odot \boldsymbol{z}$, where $\boldsymbol{\psi} = [\psi_1 \ldots \psi_l]^\top$ with $\psi_k = 1$ and $\psi_j \sim Bernoulli(\pi)$ for $j \neq k$.*

Here, the caption always mentions the feature indicating the latent class, while other features are mentioned with a probability $\pi$. Note that $\pi = 0$ corresponds to the setting where the caption is just the same as the label. The following theorem demonstrates that the model can achieve robustness only when the caption sufficiently mentions features that are not directly related to the image's latent class.

**Theorem 5.4.** *With data model 2 and $\boldsymbol{\mu}_T$ defined in Definition 5.3, let $\boldsymbol{W}_I^*$ and $\boldsymbol{W}_T^*$ be the weights of encoders trained using MMCL. Then the model's accuracy on the true distribution satisfies $Acc_{\mathscr{P}^\star}^{MMCL}(\boldsymbol{W}_I^*, \boldsymbol{W}_T^*) = 100\%$ if $\pi > \tilde{\pi}$, and $Acc_{\mathscr{P}^\star}^{MMCL}(\boldsymbol{W}_I^*, \boldsymbol{W}_T^*) \leq 50\%$ if $\pi < \tilde{\pi}$, where $\tilde{\pi} := \frac{(1+\beta)\alpha^2 - 1 + \beta}{(1-\beta)\beta^2(m-1)}$.*

As explained in Section 4.2, even if certain features do not directly indicate labels for a class, they can still help learn relationships between features (for example, not all trees are green), and this knowledge can be valuable for other classes. However, if these features are missing from the captions, they contribute less to the cross-covariance matrix used by the model for predictions (Lemma 3.2). In the extreme case where $\pi = 0$, captions reduce to labels used by SL, and robustness does not improve.

## 6 EXPERIMENTS

**A Semi-synthetic Experiment.** We conduct a carefully designed semi-synthetic binary classification experiment to showcase MMCL's robustness and the significance of rich captions. The task is to distinguish digits 0 to 4 (class 1) from digits 5 to 9 (class 2). In the training set,

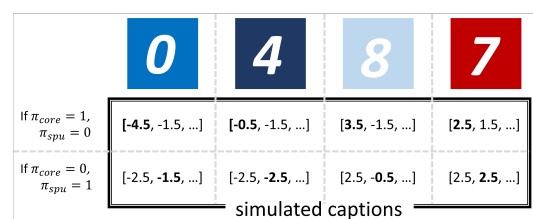

Figure 1: Construction of captions.

MNIST (Deng, 2012) digits are placed on colored backgrounds, including three types of blue and three types of red. As illustrated Figure 4, for digits 0-4, 99.5% of images have randomly selected shades of blue as the background, while the remaining 0.5% have random red backgrounds. The same applies to digits 5-9, but with blue and red swapped. In the test set, backgrounds are randomly chosen for all images. Therefore, digits represent the core feature, while colors serve as the spurious feature whose correlation with classes only exist in the training data. Captions are simulated as vectors, where the first coordinate contains digit information and the second contains color information.

Both features exhibit variance; for example, there are four variations of digits between 0 and 4 and three variations of blue backgrounds. We use $\pi_{\text{core}}$ and $\pi_{\text{spu}}$ to control the specificity of the captions, determining how much the caption mentions the variance in each feature. Being 'specific' means mentioning the exact value (e.g., specifying a particular shade of blue), while 'not specific' means referring to a value that represents an entire category (e.g., using the mean value for three shades of blue to represent any blue). Figure 1 shows an example. For more details, please refer to Appendix G.1.

We plot the OOD accuracy in Figure 2, while varying the values of $\pi_{\text{core}}$ and $\pi_{\text{spu}}$. We observe: (1) With sufficiently rich captions (high $\pi_{\text{core}}$), MMCL exhibits better robustness than SL (horizontal

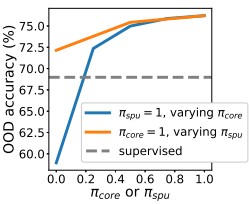

Figure 2: OOD accuracy on the semi-synthetic data. A large $\pi_{core}$ is crucial for ensuring MMCL's superior robustness compared to SL, but the value of $\pi_{spu}$ has minimal effect.

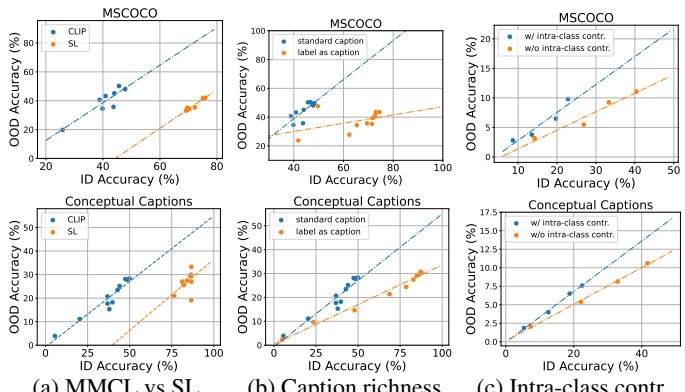

(a) MMCL vs SL    (b) Caption richness    (c) Intra-class contr.

Figure 3: (a) MMCL is more robust than SL. (b) Caption richness and (c) intra-class contrasting contribute to robustness. Note that (c) is in a different setup than (a)(b), as detailed in Appendices G.2 and G.3.

line). (2) A high $\pi_{core}$, indicating that the captions mentioning the variance of the core feature, is essential for achieving robustness, as reducing $\pi_{core}$ significantly hurts the robustness. (3) In contrast, $\pi_{spu}$ has minimal effect on robustness. It's worth noting that (2) and (3) directly validate the conclusions from Theorem 5.2. Additional discussion can be found in Appendix G.1.

**Robustness on real data.** We further coroborate our conclusions with experiments on MSCOCO (Lin et al., 2014) and Conceptual Captions (Sharma et al., 2018). We train models on these two datasets, and evaluate them on six shifted versions of ImageNets. See experimental details in Appendices G.2 and G.3).

*MMCL is more OOD-robust than SL.* We employ the widely used CLIP loss for MMCL and CE loss for SL to compare the robustness of the resulting models. Due to computational resource constraints, we adopt the simplified training setting from (Ren & Li, 2023), training a 3-layer MLP on top of frozen pretrained encoders. Following (Taori et al., 2020), we plot the ID-ODD accuracy relationship by varying the model size (width of the MLP). Fig 3a shows that models trained with MMCL demonstrate superior robustness on both datasets. Note that although the ID accuracy of CLIP is lower than that of SL, resulting in seemingly comparable OOD accuracies, we anticipate the actual advantage of CLIP to become more pronounced as the dataset size scales, similar to the original dataset used in (Radford et al., 2021). We also provide results regarding with other algorithms, including SimCLR and SupCon, and other backbone architectures in Appendix G.4.

*Richness of captions is critical in achieving robustness.* To demonstrate the impact of caption richness on robustness, we train an alternative version of CLIP wherein the captions are simplified to be the same as the labels. As depicted in Fig 3b, this modification leads to diminished robustness, which corroborates the theoretical conclusions in Section 5.

*Intra-class contrasting contributes to robustness.* To illustrate the mechanism theoretically presented in Section 4.1, we modify the CLIP loss to exclude pairs from the denominator if they are from the same class, thereby eliminating contrasting between images and texts of the same class. In this experiment, unlike the previous ones, we train the encoders from scratch and obtain different ID-OOD accuracy pairs by varying the training data size. This is because the effect of intra-class contrasting was not evident in the 3-layer MLP setting, likely due to the small model's limited capacity rendering it less sensitive to modifications in the loss. In Fig 3c, we observe that removing intra-class contrasting from the loss compromises robustness, confirming the importance of intra-class contrasting.

# 7   CONCLUSION

In this work, we provided the first theoretical explanation for MMCL's enhanced OOD robustness compared to SL. We showed conclusively that this robustness is attributed to aspects of the MMCL loss function, i.e. (1) intra-class contrasting (2) inter-class feature sharing, as well as the nature of multi-modal data i.e. (3) richness of captions. We confirmed our theoretical results using both synthetic and real-world experiments. Our findings could inspire the development of improved loss functions and data curation practices to further enhance MMCL's robustness.

**Acknowledgements** This research is partially supported by the National Science Foundation CAREER Award 2146492 and Cisco Systems.

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

## A    ADDITIONAL RELATED WORK

**Uni-modal contrastive learning.** There have been both empirical Chen et al. (2020); Chuang et al. (2020); Khosla et al. (2020) and theoretical studies about CL Wang & Isola (2020); Tosh et al. (2021a;b); Arora et al. (2019); HaoChen et al. (2021); Wen & Li (2021); Ji et al. (2021); Saunshi et al. (2022); Xue et al. (2023). We will demonstrate for the first time how the contrastive aspect of the loss can benefit OOD robustness, while noting that this advantage only exists when equipped with multi-modality and zero-shot learning, which are present in MMCL but not in unimodal CL.

## B    PRELIMINARIES

### B.1    MINIMIZER OF MMCL LOSS

Nakada et al. (2023) has shown the equivalence between minimizing linear MMCL and SVD. We reiterate this for reference in our proof. As defined in Section 3.2, our loss function is

$$\mathcal{L}_{\text{MMCL}}(\boldsymbol{W}_I,\boldsymbol{W}_T)=\frac{1}{2n(n-1)}\sum_i\sum_{j\neq i}(s_{ij}-s_{ii})+\frac{1}{2n(n-1)}\sum_i\sum_{j\neq i}(s_{ji}-s_{ii})+\frac{\rho}{2}\|\boldsymbol{W}_I^\top\boldsymbol{W}_T\|_F^2,$$

which can be rewritten as a matrix factorization objective

$$\mathcal{L}(\boldsymbol{W}_I,\boldsymbol{W}_T) = -\operatorname{Tr}(\boldsymbol{W}_I\boldsymbol{S}\boldsymbol{W}_T^\top)+\frac{\rho}{2}\|\boldsymbol{W}_I^\top\boldsymbol{W}_T\|_F^2, \tag{2}$$

where $\boldsymbol{S}$ is the cross-covariance matrix

$$\boldsymbol{S} := \frac{1}{n}\sum_i\boldsymbol{x}_{I,i}\boldsymbol{x}_{T,i}^\top - \frac{1}{n(n-1)}\sum_{i\neq j}\boldsymbol{x}_{I,i}\boldsymbol{x}_{T,j}^\top. \tag{3}$$

The following directly follows from Eckart-Young-Mirsky theorem: let $\sum_{i=1}^d \lambda_i\boldsymbol{u}_{I,i}\boldsymbol{u}_{T,i}^\top$ with $\lambda_1 \geq \lambda_2 \geq \ldots \lambda_d > 0$ be the SVD of $\boldsymbol{S}$, and let $\boldsymbol{W}_I^*, \boldsymbol{W}_T^*$ be the minimizer of the loss, then

$$\boldsymbol{W}_I^{*\top}\boldsymbol{W}_T^* = \frac{1}{\rho}\sum_{i=1}^p \lambda_i\boldsymbol{u}_{I,i}\boldsymbol{u}_{T,i}^\top \tag{4}$$

### B.2    MINIMIZER OF SL LOSS

We note that in both Data Model 1 and Data Model 2, under the assumptions we've made, the training data are separable, or separable with high probability. As shown in Soudry et al. (2018), minimizing the logistic loss or Cross-Entropy losses with a linear model at a sufficiently small step size converges to the solution for a hard-margin SVM. Therefore, in our analysis for SL, we equivalently examine this solution.

## C    ANALYSIS FOR DATA MODEL 1

### C.1    ANALYSIS FOR SL

Here, we briefly explain how Theorem 4.4 is derived from Theorem 1 in (Sagawa et al., 2020). Firstly, we provide the following version, which is a direct translation from (Sagawa et al., 2020) but with our notations. First let $n_{maj}$ denote the number of examples with $a = y$ in the training set and $n_{maj}$ denote the number of examples with $a \neq y$ in the training set. Define $\sigma_{\text{core}}'^2 := \sigma_{\text{core}}^2 + \frac{\sigma_{\xi,I}^2}{d_I}$, $\sigma_{\text{spu}}'^2 := \sigma_{\text{spu}}^2 + \frac{\sigma_{\xi,I}^2}{d_I}, \sigma_{\xi,I}'^2 := \frac{\sigma_{\xi,I}^2(d-2)}{d}$.

**Theorem C.1.** *For any* $\frac{n_{maj}}{n} \geq 1 - \frac{1}{2001}$, $\sigma_{core}'^2 \geq 1, \sigma_{spu}'^2 \leq \frac{1}{16\log 100n_{maj}}, \sigma_{\xi,I}'^2 \leq \frac{n_{maj}}{600^2}$ *and* $n_{min} \geq 100$, *there exists* $d_0$ *such that for all* $d > d_0$, *with high probability over draws of the training data*

*Test error on examples where* $a \neq y$ *achieved by* $\boldsymbol{W}^* \geq 2/3.$

It is now easy to see that the quantities in Assumption 4.2 can satisfy the conditions in the theorem above when they become sufficiently asymptotic. Note that the above statement is about test error on examples where $a \neq y$, which accounts for half of the entire distribution $\mathscr{P}^\star$, so the accuracy on the entire distribution $\mathscr{P}^\star$ is at most $\frac{1}{2} \times 100\% + \frac{1}{2} \times (1 - 2/3) = 2/3$.

## C.2  ANALYSIS FOR MMCL

We note that Theorem 4.4 holds under a more relaxed assumption, which the following anlaysis is based on.

**Assumption C.2.** *The gap between the variances of the core and spurious features is significant:* $\sigma_{core} - \sigma_{spu} = \Theta(1)$ *and* $\sigma_{spu} = O(1)$. *$p$ can be any value between $\frac{1}{2}$ and 1. We consider the high-dimensional (overparameterized) setting where $n = \omega(1)$, $d_I = \Omega(n)$ and $d_T = \Omega(n)$. The noise levels are not too large: $\sigma_{\xi,I} = O(\log n)$ and $\sigma_{\xi,T} = O(\log n)$.*

We define the following notations. We write each $\boldsymbol{z_i}$ as $\boldsymbol{z_i} = \begin{bmatrix} y_i + \zeta_{1,i} \\ a_i + \zeta_{2,i} \end{bmatrix}$, where $\zeta_{1,i}$ and $\zeta_{2,i}$ are Gaussian variables according to our definition. We also let $\boldsymbol{\zeta}_i = \begin{bmatrix} \zeta_{1,i} \\ \zeta_{2,i} \end{bmatrix}$. Let $\boldsymbol{\mathfrak{Z}} = [\boldsymbol{\zeta}_1, \boldsymbol{\zeta}_2, \ldots, \boldsymbol{\zeta}_n]$, $\Xi_I = [\boldsymbol{\xi}_{I,1}, \boldsymbol{\xi}_{I,2}, \ldots, \boldsymbol{\xi}_{I,n}]$, $\Xi_T = [\boldsymbol{\xi}_{T,1}, \boldsymbol{\xi}_{T,2}, \ldots, \boldsymbol{\xi}_{T,n}]$. Additionally, we let $\boldsymbol{F} = \begin{bmatrix} y_1, y_2, \ldots, y_n \\ a_1, a_2, \ldots, a_n \end{bmatrix}$.

### C.2.1  USEFUL CONCENTRATION BOUNDS

**Lemma C.3** (Montgomery-Smith (1990)). *Let $\{x_i\}_{i=1}^n$ be a set of random Rademacher variables, then with probability at least $1 - \delta$, the following holds*

$$|\frac{1}{n}\sum_{i=1}^n x_i| \leq \sqrt{\frac{2\ln(1/\delta)}{n}}$$

**Lemma C.4** (Mills' ratio. Exercise 6.1 in (Shorack & Shorack, 2000).). *Let $v$ be a Gaussian random variable drawn from $\mathcal{N}(0,1)$. Then for all $\lambda > 0$,*

$$\frac{\lambda}{\lambda^2 + 1}\frac{1}{\sqrt{2\pi}}e^{-\frac{\lambda^2}{2}} < \Pr(v \geq \lambda) < \frac{1}{\lambda}\frac{1}{\sqrt{2\pi}}e^{-\frac{\lambda^2}{2}}.$$

**Corollary C.5.** *Let $\{x_i\}_{i=1}^n$ be a set of random variables independently drawn from $\mathcal{N}(0,\sigma^2)$, then with probability at least $1 - O(\frac{1}{\text{poly}(n)})$, the following holds*

$$|\frac{1}{n}\sum_{i=1}^n x_i| \leq O(\sigma\sqrt{\frac{\log n}{n}})$$

**Lemma C.6** (Restatement of Theorem 6.1 from (Wainwright, 2019)). *Let $\boldsymbol{X} \in \mathbb{R}^{n \times d}$ be a random matrix with $\boldsymbol{x}_i^\top$ as its $i$-th row. Each $\boldsymbol{x}_i$ is drawn i.i.d. from $\mathcal{N}(0,\boldsymbol{\Sigma})$. Then with probability at least $1 - \delta$, the maximal singular value of $\boldsymbol{X}$, denoted as $\sigma_{\max}(\boldsymbol{X})$ satisfies the following*

$$\sigma_{\max}(\boldsymbol{X}) \leq \sqrt{n}\gamma_{\max}(\sqrt{\boldsymbol{\Sigma}})(1 + \frac{\sqrt{2\ln(1/\delta)}}{n}) + \sqrt{\text{Tr}(\boldsymbol{\Sigma})}.$$

**Lemma C.7** (Gaussian covariance estimation, Example 6.3 from (Wainwright, 2019)). *Let $\{\boldsymbol{v}_i\}_{i=1}^n$ be a set of vectors $\in \mathbb{R}^d$ independently drawn from $\mathcal{N}(0,\boldsymbol{\Sigma})$, and let $\hat{\boldsymbol{\Sigma}} = \frac{1}{n}\sum_{i=1}^n \boldsymbol{v}_i\boldsymbol{v}_i^\top$ then with probability at least $1 - 2e^{-n\delta^2/2}$*

$$\frac{\|\hat{\boldsymbol{\Sigma}} - \boldsymbol{\Sigma}\|_2}{\|\boldsymbol{\Sigma}\|_2} \leq 2\sqrt{\frac{d}{n}} + 2\delta + (\sqrt{\frac{d}{n}} + 2)^2.$$

### C.2.2  CONCENTRATION OF THE CROSS-COVARIANCE

**Concentrations in Low-dimensional Underlying Feature Space:**

**Lemma C.8.** *With probability at least $1 - \delta$ the following holds*

$$|\frac{1}{n}\sum_{i=1}^{n}a_iy_i - (2p-1)| \leq \sqrt{2\frac{\ln(1/\delta)}{n}}.$$

*Proof.* By Hoeffding's inequality. $\qquad\square$

**Lemma C.9.** *With probability at least $1 - O(\frac{1}{\mathrm{poly}(n)})$ the following holds*

$$\|\frac{1}{n}\sum_{i=1}^{n}\boldsymbol{\zeta}_i\boldsymbol{\zeta}_i^{\top} - \begin{bmatrix} \sigma_{core}^2 & 0 \\ 0 & \sigma_{spu}^2 \end{bmatrix}\|_2 \leq O(\sigma_{core}^2\sqrt{\frac{\log n}{n}})$$

*Proof.* By Lemma C.7. $\qquad\square$

**Lemma C.10.** *With probability at least $1 - O(\frac{1}{\mathrm{poly}(n)})$*

$$\|\frac{1}{n}\sum_{i=1}^{n}\begin{bmatrix} y_i \\ a_i \end{bmatrix}\boldsymbol{\zeta}_i^{\top}\|_2 \leq O(\sigma_{core}\sqrt{\frac{\log n}{n}})$$

*Proof.* By Lemma C.5. $\qquad\square$

**Lemma C.11.** *With probability at least $1 - O(\frac{1}{\mathrm{poly}(n)})$*

$$\|\frac{1}{n}\sum_{i=1}^{n}\boldsymbol{z}_i\boldsymbol{z}_i^{\top} - \begin{bmatrix} 1+\sigma_{core}^2 & 2p-1 \\ 2p-1 & 1+\sigma_{spu}^2 \end{bmatrix}\|_2 \leq O(\sqrt{\frac{\log n}{n}}) \tag{5}$$

*Proof.* Write $\frac{1}{n}\sum_{i=1}^{n}\boldsymbol{z}_i\boldsymbol{z}_i^{\top}$ as the following

$$\frac{1}{n}\sum_{i=1}^{n}\boldsymbol{z}_i\boldsymbol{z}_i^{\top} = \frac{1}{n}\sum_{i=1}^{n}\begin{bmatrix} y_i^2 & y_ia_i \\ a_iy_i & a_i^2 \end{bmatrix} + \frac{1}{n}\sum_{i=1}^{n}\begin{bmatrix} y_i \\ a_i \end{bmatrix}\boldsymbol{\zeta}_i^{\top} + \frac{1}{n}\sum_{i=1}^{n}\boldsymbol{\zeta}_i[y_i \quad a_i] + \frac{1}{n}\sum_{i=1}^{n}\boldsymbol{\zeta}_i\boldsymbol{\zeta}_i^{\top}.$$

Then invoking Lemmas C.8, C.9 and C.10 completes the proof. $\qquad\square$

**Lemma C.12.** *With probability at least $1 - O(\frac{1}{\mathrm{poly}(n)})$*

$$\|\frac{1}{n}\sum_{i=1}^{n}\boldsymbol{z}_i\|_2 \leq O(\sqrt{\frac{\log n}{n}})$$

*Proof.* Note that each of the two elements in $\frac{1}{n}\sum_{i=1}^{n}\boldsymbol{z}_i$ can be seen as the sum of the mean of $n$ independent Rademacher variables and the mean of $n$ independent Gaussian variables. Combining Lemmas C.3 and C.5 completes the proof. $\qquad\square$

**Lemma C.13.** *With probability at least $1 - O(\frac{1}{\mathrm{poly}(n)})$*

$$\left\|\frac{1}{n}\sum_{i=1}^{n}\boldsymbol{z}_i\boldsymbol{z}_i^{\top} - \frac{1}{n(n-1)}\sum_{i\neq j}^{n}\boldsymbol{z}_i\boldsymbol{z}_j^{\top} - \begin{bmatrix} 1+\sigma_{core}^2 & 2p-1 \\ 2p-1 & 1+\sigma_{spu}^2 \end{bmatrix}\right\|_2 = O(\sqrt{\frac{\log n}{n}}).$$

*Proof.*

$$\frac{1}{n}\sum_{i=1}^{n}\boldsymbol{z}_i\boldsymbol{z}_i^{\top} - \frac{1}{n(n-1)}\sum_{i\neq j}^{n}\boldsymbol{z}_i\boldsymbol{z}_j^{\top} = \frac{1}{n}\sum_{i=1}^{n}\boldsymbol{z}_i\boldsymbol{z}_i^{\top} - \frac{1}{n(n-1)}\sum_{i=1}^{n}\boldsymbol{z}_i\sum_{j=1}^{n}\boldsymbol{z}_j + \frac{1}{n(n-1)}\sum_{i=1}^{n}\boldsymbol{z}_i\boldsymbol{z}_i^{\top}$$

$$= \frac{1}{n}\sum_{i=1}^{n}\boldsymbol{z}_i\boldsymbol{z}_i^{\top} - \frac{n}{n-1}\frac{1}{n}\sum_{i=1}^{n}\boldsymbol{z}_i\frac{1}{n}\sum_{j=1}^{n}\boldsymbol{z}_j + \frac{1}{n(n-1)}\sum_{i=1}^{n}\boldsymbol{z}_i\boldsymbol{z}_i^{\top}.$$

Note that by Lemma C.12, the norm of the second term on the RHS is $O(\frac{\log n}{n})$, and by Lemma C.11 the norm of the third term is $O(\frac{1}{n})$. Combining these results and applying Lemma C.11 to the first term yields:

$$\left\| \frac{1}{n} \sum_{i=1}^{n} z_i z_i^\top - \frac{1}{n(n-1)} \sum_{i \neq j}^{n} z_i z_j^\top - \begin{bmatrix} 1 + \sigma_{\text{core}}^2 & 2p - 1 \\ 2p - 1 & 1 + \sigma_{\text{spu}}^2 \end{bmatrix} \right\|_2 = O\left(\sqrt{\frac{\log n}{n}}\right).$$

$\square$

**Concentrations in High-dimensional Input Space:**

**Corollary C.14.** *By applying Lemma C.6, we can conclude that the following statements hold with a probability of at least $1 - 3\delta$*

$$\|\mathbf{3}\|_2 \leq \sqrt{n}\sigma_{core}\left(1 + \frac{\sqrt{2\ln(1/\delta)}}{n}\right) + \sqrt{\sigma_{core}^2 + \sigma_{spu}^2}$$

$$\|\Xi_I\|_2 \leq \sigma_{\xi,I}\left(\sqrt{\frac{n}{d_I}} + \sqrt{\frac{2\ln(1/\delta)}{nd_I}}\right) + \sigma_{\xi,I}$$

$$\|\Xi_T\|_2 \leq \sigma_{\xi,T}\left(\sqrt{\frac{n}{d_T}} + \sqrt{\frac{2\ln(1/\delta)}{nd_T}}\right) + \sigma_{\xi,T}$$

**Lemma C.15.** *With probability at least $1 - O(\frac{1}{\text{poly}(n)})$*

$$\|\frac{1}{n} \sum_{i=1}^{n} \boldsymbol{\xi}_{I,i}\| \leq O\left(\frac{\sigma_{\xi,I}}{\sqrt{n}}\right) \quad and \quad \|\frac{1}{n} \sum_{i=1}^{n} \boldsymbol{\xi}_{T,i}\| \leq O\left(\frac{\sigma_{\xi,T}}{\sqrt{n}}\right).$$

*Proof.* This can be obtained by recognizing that $\frac{1}{n}\sum_{i=1}^{n} \boldsymbol{\xi}_{I,i}$ can be treated as a single sample from $\mathcal{N}(0, \frac{\sigma_I^2}{nd_I}\mathbf{I}_{d_I})$ and by applying Lemma C.6 with $\delta = \frac{1}{\text{poly}(n)}$. A similar argument applies to $\frac{1}{n}\sum_{i=1}^{n} \boldsymbol{\xi}_{T,i}$. $\square$

**Lemma C.16.** *With probability at least $1 - O(\frac{1}{\text{poly}(n)})$*

$$\left\| \boldsymbol{S} - \boldsymbol{D}_I \begin{bmatrix} 1 + \sigma_{core}^2 & 2p - 1 \\ 2p - 1 & 1 + \sigma_{spu}^2 \end{bmatrix} \boldsymbol{D}_T^\top \right\|_2 \leq O\left(\frac{\sigma_{\xi,T} + \sigma_{\xi,I}}{\sqrt{n}} + \frac{\sigma_{\xi,I}\sigma_{\xi,T}}{n}\right)$$

$$+ O\left(\frac{\sigma_{\xi,A}\sqrt{\log n} + \sigma_{\xi,B}\sqrt{\log n} + \sigma_{\xi,I}\sigma_{\xi,T}}{n}\right)$$

$$+ O\left(\sqrt{\frac{\log n}{n}}\right),$$

*where $\boldsymbol{S}$ is defined in Equation 3.*

*Proof.* Firstly let's write $\boldsymbol{S}$ as

$$\boldsymbol{S} = \frac{1}{n} \sum_{i=1}^{n} \boldsymbol{x}_{I,i} \boldsymbol{x}_{T,i}^\top - \frac{1}{n(n-1)} \sum_{i \neq j} \boldsymbol{x}_{I,i} \boldsymbol{x}_{T,i}^\top \tag{6}$$

$$= \frac{1}{n} \sum_{i=1}^{n} \boldsymbol{D}_I z_i z_i^\top \boldsymbol{D}_T^\top - \frac{1}{n(n-1)} \sum_{i \neq j} \boldsymbol{D}_I z_i z_j^\top \boldsymbol{D}_T^\top + \boldsymbol{R}$$

where

$$\boldsymbol{R} = \frac{1}{n} \sum_{i=1}^{n} (\boldsymbol{D}_I \boldsymbol{z}_i \boldsymbol{\xi}_{T,i}^\top + \boldsymbol{\xi}_{I,i} \boldsymbol{z}_i^\top \boldsymbol{D}_T^\top + \boldsymbol{\xi}_{I,i} \boldsymbol{\xi}_{T,i}^\top) - \frac{1}{n(n-1)} \sum_{i \neq j} (\boldsymbol{D}_I \boldsymbol{z}_i \boldsymbol{\xi}_{T,j}^\top + \boldsymbol{\xi}_{I,i} \boldsymbol{z}_j^\top \boldsymbol{D}_T^\top + \boldsymbol{\xi}_{I,i} \boldsymbol{\xi}_{T,j}^\top)$$

$$= \underbrace{\frac{1}{n-1} \sum_{i=1}^{n} (\boldsymbol{D}_I \boldsymbol{z}_i \boldsymbol{\xi}_{T,i}^\top + \boldsymbol{\xi}_{I,i} \boldsymbol{z}_i^\top \boldsymbol{D}_T^\top + \boldsymbol{\xi}_{I,i} \boldsymbol{\xi}_{T,i}^\top)}_{\boldsymbol{R}_1}$$

$$- \underbrace{\frac{1}{n(n-1)} \sum_{i=1}^{n} \sum_{j=1}^{n} (\boldsymbol{D}_I \boldsymbol{z}_i \boldsymbol{\xi}_{T,j}^\top + \boldsymbol{\xi}_{I,i} \boldsymbol{z}_j^\top \boldsymbol{D}_T^\top + \boldsymbol{\xi}_{I,i} \boldsymbol{\xi}_{T,j}^\top)}_{\boldsymbol{R}_2}.$$

Let's rewrite $\boldsymbol{R}_1$ as

$$\boldsymbol{R}_1 = \frac{1}{n-1} \left( \boldsymbol{D}_I (\boldsymbol{F} + \boldsymbol{3}) \Xi_T^\top + \Xi_I (\boldsymbol{F} + \boldsymbol{3})^\top \boldsymbol{D}_T^\top + \Xi_I \Xi_T^\top \right).$$

Then

$$\|\boldsymbol{R}_1\|_2 \leq \frac{1}{n-1} \left( \|\boldsymbol{D}_I\|_2 (\|\boldsymbol{F}\|_2 + \|\boldsymbol{3}\|_2) \|\Xi_T\|_2 + \|\Xi_I\|_2 (\|\boldsymbol{F}\|_2 + \|\boldsymbol{3}\|_2) \|\boldsymbol{D}_T\|_2 + \|\Xi_I\|_2 \|\Xi_T\|_2 \right).$$

Note that $\|\boldsymbol{D}_I\|_2 = \|\boldsymbol{D}_T\|_2 = 1$ since they have orthonormal columns. Additionally, we can observe that $\|\boldsymbol{F}\|_2 \leq \|\boldsymbol{F}\|_F = \sqrt{2n}$. By combining these and applying Corollary C.14 with $\delta = O(\frac{1}{\text{poly}(n)})$ we obtain the following

$$\|\boldsymbol{R}_1\|_2 \leq O\left( \frac{\sigma_{\xi,T} + \sigma_{\text{core}} \sigma_{\xi,T} + \sigma_{\xi,I} + \sigma_{\text{core}} \sigma_{\xi,I}}{\sqrt{n}} + \frac{\sigma_{\xi,I} \sigma_{\xi,T}}{n} \right)$$

$$= O\left( \frac{\sigma_{\xi,T} + \sigma_{\xi,I}}{\sqrt{n}} + \frac{\sigma_{\xi,I} \sigma_{\xi,T}}{n} \right) \tag{7}$$

Next, we rewrite $\boldsymbol{R}_2$ as

$$\boldsymbol{R}_2 = \frac{n}{n-1} \left( \boldsymbol{D}_I \frac{1}{n} \sum_{i=1}^{n} \boldsymbol{z}_i \left( \frac{1}{n} \sum_{j=1}^{n} \boldsymbol{\xi}_{T,j} \right)^\top + \frac{1}{n} \sum_{i=1}^{n} \boldsymbol{\xi}_{I,i} \left( \frac{1}{n} \sum_{j=1}^{n} \boldsymbol{z}_j \right)^\top \boldsymbol{D}_T^\top + \frac{1}{n} \sum_{i=1}^{n} \boldsymbol{\xi}_{I,i} \left( \frac{1}{n} \sum_{i=1}^{n} \boldsymbol{\xi}_{T,i} \right)^\top \right).$$

Then applying Lemmas C.12 and C.15 yields

$$\|\boldsymbol{R}_2\|_2 \leq O\left( \frac{\sigma_{\xi,A} \sqrt{\log n} + \sigma_{\xi,B} \sqrt{\log n} + \sigma_{\xi,I} \sigma_{\xi,T}}{n} \right). \tag{8}$$

Additionally we observe that

$$\frac{1}{n} \sum_{i=1}^{n} \boldsymbol{D}_I \boldsymbol{z}_i \boldsymbol{z}_i^\top \boldsymbol{D}_T^\top - \frac{1}{n(n-1)} \sum_{i \neq j} \boldsymbol{D}_I \boldsymbol{z}_i \boldsymbol{z}_j^\top \boldsymbol{D}_T^\top = \boldsymbol{D}_I \left( \frac{1}{n} \sum_{i=1}^{n} \boldsymbol{z}_i \boldsymbol{z}_i^\top - \frac{1}{n(n-1)} \sum_{i \neq j} \boldsymbol{z}_i \boldsymbol{z}_j^\top \right) \boldsymbol{D}_T^\top.$$

Therefore by Lemma C.11

$$\left\| \frac{1}{n} \sum_{i=1}^{n} \boldsymbol{D}_I \boldsymbol{z}_i \boldsymbol{z}_i^\top \boldsymbol{D}_T^\top - \frac{1}{n(n-1)} \sum_{i \neq j} \boldsymbol{D}_I \boldsymbol{z}_i \boldsymbol{z}_j^\top \boldsymbol{D}_T^\top - \boldsymbol{D}_I \begin{bmatrix} 1 + \sigma_{\text{core}}^2 & 0 \\ 0 & 1 + \sigma_{\text{spu}}^2 \end{bmatrix} \boldsymbol{D}_T^\top \right\|_2$$

$$\leq \|\boldsymbol{D}_I\|_2 \|\boldsymbol{D}_T\|_2 O\left( \sqrt{\frac{\log n}{n}} \right)$$

$$= O\left( \sqrt{\frac{\log n}{n}} \right) \tag{9}$$

Then, combining Equations 9, 7, 8, 6 yields

$$\left\| \boldsymbol{S} - \boldsymbol{D}_I \begin{bmatrix} 1 + \sigma_{\text{core}}^2 & 2p - 1 \\ 2p - 1 & 1 + \sigma_{\text{spu}}^2 \end{bmatrix} \boldsymbol{D}_T^\top \right\|_2 \leq O\left( \frac{\sigma_{\xi,T} + \sigma_{\xi,I}}{\sqrt{n}} + \frac{\sigma_{\xi,I} \sigma_{\xi,T}}{n} \right)$$

$$+ O\left( \frac{\sigma_{\xi,A} \sqrt{\log n} + \sigma_{\xi,B} \sqrt{\log n} + \sigma_{\xi,I} \sigma_{\xi,T}}{n} \right)$$

$$+ O\left( \sqrt{\frac{\log n}{n}} \right)$$

$\square$

### C.2.3 PERTURBATION IN SVD

**Lemma C.17.** $\boldsymbol{G}^* \in \mathbb{R}^{d_I \times d_T}$ *is a matrix whose SVD is* $\sum_{i=1}^{2} \lambda_i \boldsymbol{u}_i \boldsymbol{v}_i^\top$ *where* $\lambda_1 > \lambda_2 > 0$, $\lambda_1 = \Theta(1)$, $\lambda_2 = \Theta(1)$ *and* $\lambda_1 - \lambda_2 = \Theta(1)$. $\boldsymbol{G} = \boldsymbol{G}^* + \boldsymbol{E}$ *where* $\|\boldsymbol{E}\|_2 \leq \epsilon$. *Let* $\sum_{i=1}^{r} \tilde{\lambda}_i \tilde{\boldsymbol{u}}_i \tilde{\boldsymbol{v}}_i^\top$ *be the SVD of* $\boldsymbol{G}$. *If* $\epsilon = o(1)$, *then* $\|\sum_{i=1}^{p} \tilde{\lambda}_i \tilde{\boldsymbol{u}}_i \tilde{\boldsymbol{v}}_i^\top - \boldsymbol{G}^*\|_2 \leq O(\sqrt{\epsilon})$, *where* $2 \leq p \leq r$.

*Proof.* Weyl's Theorem tells us that $|\tilde{\lambda}_1 - \lambda_1| \leq \epsilon$, $|\tilde{\lambda}_2 - \lambda_2| \leq \epsilon$ and $|\tilde{\lambda}_i| \leq \epsilon$ for $i \geq 3$. Now, let's define $\delta = \min\{\lambda_1 - \lambda_2, \lambda_2\}$. By applying Wedin's Theorem (Wedin, 1972) with the singular values partitioned into $\{\lambda_1\}$ and $\{\lambda_i\}_{i \neq 1}$, we have $\sin\theta(\boldsymbol{u}_1, \tilde{\boldsymbol{u}}_1) \leq O(\frac{\epsilon}{\delta})$. Similarly, by applying Wedin's Theorem with the singular values partitioned into $\{\lambda_2\}$ and $\{\lambda_i\}_{i \neq 2}$, we have $\sin\theta(\boldsymbol{u}_2, \tilde{\boldsymbol{u}}_2) \leq O(\frac{\epsilon}{\delta})$. Considering that $\delta = \Theta(1)$, we further have $\sin\theta(\boldsymbol{u}_1, \tilde{\boldsymbol{u}}_1) \leq O(\epsilon)$ and $\sin\theta(\boldsymbol{u}_2, \tilde{\boldsymbol{u}}_2) \leq O(\epsilon)$. Similar conclusions hold for $\tilde{\boldsymbol{v}}_i$'s as well.

Now, for $i = 1, 2$, let's examine the difference between $\tilde{\boldsymbol{u}}_i$ and $\boldsymbol{u}_i$:

$$\begin{aligned}
\|\tilde{\boldsymbol{u}}_i - \boldsymbol{u}_i\|^2 &= 2(1 - \cos\theta(\boldsymbol{u}_i, \tilde{\boldsymbol{u}}_i)) \\
&= 2(1 - \sqrt{1 - \sin\theta(\boldsymbol{u}_i, \tilde{\boldsymbol{u}}_i)}) \\
&= 2\frac{\sin\theta(\boldsymbol{u}_i, \tilde{\boldsymbol{u}}_i)}{1 + \sqrt{1 - \sin\theta(\boldsymbol{u}_i, \tilde{\boldsymbol{u}}_i)}} \\
&= O(\sin\theta(\boldsymbol{u}_i, \tilde{\boldsymbol{u}}_i)) \\
&\leq O(\epsilon).
\end{aligned}$$

Therefore, $\|\tilde{\boldsymbol{u}}_i - \boldsymbol{u}_i\| \leq O(\sqrt{\epsilon})$. Similarly, we can deduce that $\|\tilde{\boldsymbol{v}}_i - \boldsymbol{v}_i\| \leq O(\sqrt{\epsilon})$. By some algebraic calculations, we further have $\|\sum_{i=1}^{2} \tilde{\lambda}_i \tilde{\boldsymbol{u}}_i \tilde{\boldsymbol{v}}_i^\top - \boldsymbol{G}^*\|_2 \leq O(\sqrt{\epsilon})$. Now, if $p \geq 3$, we previously established that $|\tilde{\lambda}_i| \leq \epsilon$ for $i \geq 3$. Consequently, $\|\sum_{i=3}^{p} \tilde{\lambda}_i \tilde{\boldsymbol{u}}_i \tilde{\boldsymbol{v}}_i^\top\|_2 = \max\{\tilde{\lambda}_i\}_{i=3}^{p} \leq \epsilon$. Thus, $\|\sum_{i=1}^{p} \tilde{\lambda}_i \tilde{\boldsymbol{u}}_i \tilde{\boldsymbol{v}}_i^\top - \boldsymbol{G}^*\|_2 \leq \|\sum_{i=1}^{2} \tilde{\lambda}_i \tilde{\boldsymbol{u}}_i \tilde{\boldsymbol{v}}_i^\top - \boldsymbol{G}^*\|_2 + \|\sum_{i=3}^{p} \tilde{\lambda}_i \tilde{\boldsymbol{u}}_i \tilde{\boldsymbol{v}}_i^\top\|_2 \leq O(\sqrt{\epsilon})$. $\square$

For convenience, let's define

$$\epsilon_0 := \frac{\sigma_{\xi,T} + \sigma_{\xi,I}}{\sqrt{n}} + \frac{\sigma_{\xi,I}\sigma_{\xi,T}}{n} + \frac{\sigma_{\xi,A}\sqrt{\log n} + \sigma_{\xi,B}\sqrt{\log n} + \sigma_{\xi,I}\sigma_{\xi,T}}{n} + \sqrt{\frac{\log n}{n}}.$$

**Corollary C.18.** *The minimizer satisfies the following with a probability of at least* $1 - O(\frac{1}{\text{poly}(n)})$,

$$\|\boldsymbol{W}_I^{*\top}\boldsymbol{W}_T^* - \frac{1}{\rho}\boldsymbol{D}_I \begin{bmatrix} 1 + \sigma_{core}^2 & 2p-1 \\ 2p-1 & 1 + \sigma_{spu}^2 \end{bmatrix} \boldsymbol{D}_T^\top\|_2 \leq \frac{1}{\rho}O(\sqrt{\epsilon_0}).$$

### C.2.4 ZERO-SHOT CLASSIFICATION

In the following analysis, we will examine the zero-shot accuracy in an event where

$$\|\boldsymbol{W}_I^{*\top}\boldsymbol{W}_T^* - \frac{1}{\rho}\boldsymbol{D}_I \begin{bmatrix} 1 + \sigma_{\text{core}}^2 & 2p-1 \\ 2p-1 & 1 + \sigma_{\text{spu}}^2 \end{bmatrix} \boldsymbol{D}_T^\top\|_2 \leq \frac{1}{\rho}O(\sqrt{\epsilon_0}).$$

It is important to note that such an event occurs with a probability of at least $1 - O(\frac{1}{\text{poly}(n)})$ by Corollary C.18.

Let $\boldsymbol{x}_I = \boldsymbol{D}_I \begin{bmatrix} y + \zeta_1 \\ a + \zeta_2 \end{bmatrix} + \boldsymbol{\xi}_I$ be a test input satisfies $y = 1, a = -1$.

By Lemma C.6, with probability at least $1 - O(\frac{1}{\text{poly}(n)})$

$$\|\boldsymbol{x}_I\| \leq O(\log n). \tag{10}$$

Recall that the prompts are $\boldsymbol{p}_y = \boldsymbol{D}_T \begin{bmatrix} y \\ 0 \end{bmatrix}$ for $y = -1, 1$. Then

$$\left| \boldsymbol{x}_I^\top \boldsymbol{W}_I^{*\top} \boldsymbol{W}_T^* \boldsymbol{x}_T^{(y)} - \frac{1}{\rho} \boldsymbol{x}_I^\top \boldsymbol{D}_I \begin{bmatrix} 1 + \sigma_{\text{core}}^2 & 2p - 1 \\ 2p - 1 & 1 + \sigma_{\text{spu}}^2 \end{bmatrix} \boldsymbol{D}_T^\top \boldsymbol{x}_T^{(y)} \right| \leq \|\boldsymbol{x}_I\| \|\boldsymbol{x}_T^{(y)}\| \frac{1}{\rho} O(\sqrt{\epsilon_0}) \qquad (11)$$

$$\leq \frac{1}{\rho} O(\sqrt{\epsilon_0} \log n).$$

Now let's look at $\boldsymbol{x}_I^\top \boldsymbol{D}_I \begin{bmatrix} 1 + \sigma_{\text{core}}^2 & 2p - 1 \\ 2p - 1 & 1 + \sigma_{\text{spu}}^2 \end{bmatrix} \boldsymbol{D}_T^\top \boldsymbol{x}_T^{(y)}$.

$$\boldsymbol{x}_I^\top \boldsymbol{D}_I \begin{bmatrix} 1 + \sigma_{\text{core}}^2 & 2p - 1 \\ 2p - 1 & 1 + \sigma_{\text{spu}}^2 \end{bmatrix} \boldsymbol{D}_T^\top \boldsymbol{x}_T^{(y)} = y \left( (1 + \zeta_1)(1 + \sigma_{\text{core}}^2) + (-1 + \zeta_2)(2p - 1) \right). \qquad (12)$$

In order for the model to make correct predictions for this example, we need $\boldsymbol{x}_I^\top \boldsymbol{W}_I^{*\top} \boldsymbol{W}_T^* \boldsymbol{x}_T^{(1)} > \boldsymbol{x}_I^\top \boldsymbol{W}_I^{*\top} \boldsymbol{W}_T^* \boldsymbol{x}_T^{(-1)}$. Based on Equation 11, we can establish the following sufficient condition:

$$\frac{1}{\rho} \boldsymbol{x}_I^\top \boldsymbol{D}_I \begin{bmatrix} 1 + \sigma_{\text{core}}^2 & 2p - 1 \\ 2p - 1 & 1 + \sigma_{\text{spu}}^2 \end{bmatrix} \boldsymbol{D}_T^\top \boldsymbol{x}_T^{(1)} - \frac{1}{\rho} O(\sqrt{\epsilon_0} \log n)$$

$$> \frac{1}{\rho} \boldsymbol{x}_I^\top \boldsymbol{D}_I \begin{bmatrix} 1 + \sigma_{\text{core}}^2 & 2p - 1 \\ 2p - 1 & 1 + \sigma_{\text{spu}}^2 \end{bmatrix} \boldsymbol{D}_T^\top \boldsymbol{x}_T^{(-1)} + \frac{1}{\rho} O(\sqrt{\epsilon_0} \log n).$$

By substituting Equation 12 into the above expressions, we obtain:

$$(1 + \zeta_1)(1 + \sigma_{\text{core}}^2) + (-1 + \zeta_2)(2p - 1) - O(\sqrt{\epsilon_0} \log n) > 0. \qquad (13)$$

Let $\epsilon_1$ denote last term on the LHS, i.e., $\epsilon_1 = O(\sqrt{\epsilon_0} \log n)$.

By recognizing that $\zeta_1(1 + \sigma_{\text{core}}^2) + \zeta_2(2p - 1)$ is a variable follows the Gaussian distribution $\mathcal{N}\left(0, (1 + \sigma_{\text{core}}^2)^2 \sigma_{\text{core}}^2 + (2p - 1)^2 \sigma_{\text{spu}}^2\right)$, we can derive the following probability:

$$\Pr\left((1 + \zeta_1)(1 + \sigma_{\text{core}}^2) + (-1 + \zeta_2)(2p - 1) - \epsilon_1 > 0\right) \qquad (14)$$

$$= \Pr_{v \sim \mathcal{N}(0,1)}\left(v > \frac{2p - 2 - \sigma_{\text{core}}^2 + \epsilon_1}{\sqrt{(1 + \sigma_{\text{core}}^2)^2 \sigma_{\text{core}}^2 + (2p - 1)^2 \sigma_{\text{spu}}^2}}\right)$$

$$= 1 - \boldsymbol{\Phi}\left(\frac{2p - 2 - \sigma_{\text{core}}^2 + \epsilon_1}{\sqrt{(1 + \sigma_{\text{core}}^2)^2 \sigma_{\text{core}}^2 + (2p - 1)^2 \sigma_{\text{spu}}^2}}\right),$$

where $\boldsymbol{\Phi}(\cdot)$ denotes the CDF of the standard normal distribution.

Therefore, we can conclude that, in order for the model to make correct predictions, the failure probability is bounded by $\boldsymbol{\Phi}\left(\frac{2p - 2 - \sigma_{\text{core}}^2 + \epsilon_1}{\sqrt{(1 + \sigma_{\text{core}}^2)^2 \sigma_{\text{core}}^2 + (2p - 1)^2 \sigma_{\text{spu}}^2}}\right)$ plus the probability for Equation 10 to not hold. Thus, the error rate on test examples where $y = 1, a = -1$, denoted by $\mathbf{Err}_{(1,-1)}$, is bounded by:

$$\mathbf{Err}_{(1,-1)}(\boldsymbol{W}_I^*, \boldsymbol{W}_T^*)$$

$$\leq \boldsymbol{\Phi}\left(\frac{2p - 2 - \sigma_{\text{core}}^2 + \epsilon_1}{\sqrt{(1 + \sigma_{\text{core}}^2)^2 \sigma_{\text{core}}^2 + (2p - 1)^2 \sigma_{\text{spu}}^2}}\right) + O\left(\frac{1}{\text{poly}(n)}\right)$$

$$= \boldsymbol{\Phi}\left(\frac{2p - 2 - \sigma_{\text{core}}^2}{\sqrt{(1 + \sigma_{\text{core}}^2)^2 \sigma_{\text{core}}^2 + (2p - 1)^2 \sigma_{\text{spu}}^2}}\right) + O\left(\frac{\epsilon_1}{\sqrt{(1 + \sigma_{\text{core}}^2)^2 \sigma_{\text{core}}^2 + (2p - 1)^2 \sigma_{\text{spu}}^2}} + \frac{1}{\text{poly}(n)}\right) \qquad ①$$

$$= \boldsymbol{\Phi}\left(\frac{2p - 2 - \sigma_{\text{core}}^2}{\sqrt{(1 + \sigma_{\text{core}}^2)^2 \sigma_{\text{core}}^2 + (2p - 1)^2 \sigma_{\text{spu}}^2}}\right) + O\left(\epsilon_1 + \frac{1}{\text{poly}(n)}\right).$$

Note that Equation ① is obtained by taking the first order Taylor approximation for $\boldsymbol{\Phi}$.

We can also derive that $\mathbf{Err}_{(-1,1)}$ is bounded in the same way as above. Similarly, we can obtain

$$\mathbf{Err}_{(1,1)} = \mathbf{Err}_{(-1,-1)} \leq \Phi(\frac{-2p - \sigma_{\text{core}}^2}{\sqrt{(1 + \sigma_{\text{core}}^2)^2 \sigma_{\text{core}}^2 + (2p-1)^2 \sigma_{\text{spu}}^2}}) + O\left(\epsilon_1 + \frac{1}{\text{poly}(n)}\right). \quad (15)$$

Converting error rate to accuracy yields Theorem 4.4. Note that the error rate can be lower bounded with the same non-negligible term. For example, $\mathbf{Err}_{(-1,-1)} \geq \Phi(\frac{-2p-\sigma_{\text{core}}^2}{\sqrt{(1+\sigma_{\text{core}}^2)^2\sigma_{\text{core}}^2+(2p-1)^2\sigma_{\text{spu}}^2}}) - o(1)$.

### C.3 MMCL WITH FEATURE MASKING

With feature masking, the proof is almost the same as above, but we just need to realize that the only change is in the covariance of features, thus we would get

$$\|\boldsymbol{W}_I^{*\top}\boldsymbol{W}_T^* - \frac{1}{\rho}\boldsymbol{D}_I \begin{bmatrix} 1 + \pi_{\text{core}}\sigma_{\text{core}}^2 & 2p - 1 \\ 2p - 1 & 1 + \pi_{\text{spu}}\sigma_{\text{spu}}^2 \end{bmatrix} \boldsymbol{D}_T^\top\|_2 \leq \frac{1}{\rho}O(\sqrt{\epsilon_0}).$$

Then going through the same steps as in Section C.2.4 yields Theorem 5.2.

## D ANALYSIS FOR DATA MODEL 2

### D.1 ANALYSIS FOR SL

For a vector $\boldsymbol{v}$, we use $\boldsymbol{v}[j]$ to denote its $j$-th element. W.O.L.G., let $\boldsymbol{D}_I \in \text{d}_I \times l$ be the first $l$-th columns of an identity matrix (since by definition $\boldsymbol{D}_I$ has orthonormal columns and we can always apply a change of basis).

By definition, the solution of SVM, denoted by $\boldsymbol{W}^*$, satisfies

$$\boldsymbol{W}^* = \arg\min_{\boldsymbol{W}=[\boldsymbol{w}_1 \ \dots \ \boldsymbol{w}_{2m}]} \|\boldsymbol{W}\| \ s.t. \ \boldsymbol{w}_y^\top \boldsymbol{x}_I - \boldsymbol{w}_{y'}^\top \boldsymbol{x}_I \geq 1, \ \forall(\boldsymbol{x}_I, y) \text{ and } y' \neq y \text{ in the training set.}$$

$$(16)$$

Construct $\hat{\boldsymbol{W}} = [\hat{\boldsymbol{w}}_1 \ \hat{\boldsymbol{w}}_2 \ \dots \ \hat{\boldsymbol{w}}_{2m}]$, where $\hat{\boldsymbol{w}}_{2k-1+(1+c)/2}$'s $k$-th element is $\frac{c}{(1-\beta)(1+\alpha^2)}$, its $(k+m)$-th element is $\frac{c\alpha}{(1-\beta)(1+\alpha^2)}$ and its other elements are zero. It is easy to check that $\hat{\boldsymbol{W}}$ satisfies the condition $\hat{\boldsymbol{w}}_y^\top \boldsymbol{x}_I - \hat{\boldsymbol{w}}_{y'}^\top \boldsymbol{x}_I \geq 1$, for any $(\boldsymbol{x}_I, y)$ and $y' \neq y$ from the training set, and $\|\hat{\boldsymbol{W}}\| = \sqrt{2m} \times \frac{1}{(1-\beta)(1+\alpha^2)} \times \sqrt{1+\alpha^2}$.

**Definition D.1.** *Define $\mathcal{S}$ as the set of classes such that $\forall(k,c) \in \mathcal{S}$ ($(k,c)$ is the alias of $y$), the following holds: $\exists$ an example $(\boldsymbol{x}_I, y)$ from $\mathscr{P}^\star$, such that $\boldsymbol{x}_I$'s $(k+m)$-th element is $-\alpha c$, and the margin maximizer on training data can make the correct prediction on this example, i.e.,*

$$[\boldsymbol{x}_I]_{k+m} = -\alpha c \quad \text{and} \quad \arg\max_j \hat{\boldsymbol{w}}_j^\top \boldsymbol{x}_I = y. \quad (17)$$

Define $\epsilon := |\mathcal{S}|/m$ Next, we are going to show that the assumption $\epsilon > \frac{8}{(1+\alpha^2)(1-\beta)^2-8}$ would lead to contradiction.

Assume $|\mathcal{S}| \geq \epsilon m$. For each $(k,c) \in \mathcal{S}$, let $\boldsymbol{x}_I^+$ be the input of an example satisfying the condition in equation 17. Then $\boldsymbol{x}_I^+$'s $k$-th element is $c$, and $(k+m)$-th element is $-\alpha c$. We construct an example $\boldsymbol{x}_I^-$ whose $k$-th element is $c$, $(k+m)$-th element is $\alpha c$, and the remaining elements are the opposite of the corresponding elements in $\boldsymbol{x}_I^+$.

From assumption 17, we have

$$\forall j \neq y, \boldsymbol{w}_y^{*\top}\boldsymbol{x}_I^+ - \boldsymbol{w}_j^{*\top}\boldsymbol{x}_I^+ > 0. \quad (18)$$

Note that $\boldsymbol{x}_I^-$ shows in the training set. By the condition for the SVM solution, we have

$$\forall j \neq y, \boldsymbol{w}_y^{*\top}\boldsymbol{x}_I^- - \boldsymbol{w}_j^{*\top}\boldsymbol{x}_I^- \geq 1. \quad (19)$$

Now, for any $j \neq y$, let's compute a lower bound for $\|\boldsymbol{w}_y^* - \boldsymbol{w}_j^*\|$. Any vector $\boldsymbol{w}_y^* - \boldsymbol{w}_j^*$ can be written as $a\boldsymbol{x}_I^+ + b\boldsymbol{x}_I^- + \boldsymbol{v}_\perp$, where $\boldsymbol{v}_\perp$ is a vector orthogonal to both $\boldsymbol{x}_I^+$ and $\boldsymbol{x}_I^-$. By equations 18 and 19, we have

$$\begin{cases} c_1 a + c_2 b > 0 \\ c_2 a + c_3 b \geq 1 \end{cases}, \tag{20}$$

where

$$c_1 = \|\boldsymbol{x}_I^+\|^2 = (1 + \alpha^2)\left(1 + (m-1)\beta^2\right) \tag{21}$$

$$c_2 = \boldsymbol{x}_I^{+\top} \boldsymbol{x}_I^- = 1 - \alpha^2 - (m-1)\beta^2(1 + \alpha^2) \tag{22}$$

$$c_3 = \|\boldsymbol{x}_I^-\|^2 = (1 + \alpha^2)\left(1 + (m-1)\beta^2\right). \tag{23}$$

Remember that we want to lower bound the following quantity

$$\|\boldsymbol{w}_y^* - \boldsymbol{w}_j^*\|^2 = c_1 a^2 + c_3 b^2 + 2c_2 ab + \|\boldsymbol{v}_\perp\|^2, \tag{24}$$

given the constraints in 20. By equations 21 to 23, $(2c_2)^2 - 4c_1 c_3 < 0$. Then $c_1 a^2 + c_3 b^2 + 2c_2 ab = D$ is always an ellipse or a circle centered at the origin in the a-b coordinate system, where a larger $D$ means a larger radius. Also, given that $c_1 > 0, c_2 < 0, c_3 > 0$, by plotting the feasible area, we can observe that $D$ achieves its minimum when the intersection point of lines $c_1 a + c_2 b = 0$ and $c_2 a + c_3 b = 1$ is exactly at the ellipse (or circle). Now we can solve for the minimum of $c_1 a^2 + c_3 b^2 + 2c_2 ab$:

$$c_1 a^2 + c_3 b^2 + 2c_2 ab \geq \frac{c_1}{c_1 c_3 - c_2^2}.$$

Then by equation 24 we have

$$\|\boldsymbol{w}_y^* - \boldsymbol{w}_j^*\|^2 \geq \frac{c_1}{c_1 c_3 - c_2^2}.$$

Then we get the following lower bound for $\|\boldsymbol{w}_y^*\|^2 + \|\boldsymbol{w}_j^*\|^2$

$$\begin{aligned} \|\boldsymbol{w}_y^*\|^2 + \|\boldsymbol{w}_j^*\|^2 &\geq \frac{(\|\boldsymbol{w}_y^*\| + \|\boldsymbol{w}_j^*\|)^2}{2} \\ &\geq \frac{(\|\boldsymbol{w}_y^* - \boldsymbol{w}_j^*\|)^2}{2} \\ &\geq \frac{1}{2} \frac{c_1}{c_1 c_3 - c_2^2} \\ &= \frac{1}{8} \frac{(1 + \alpha^2)\left(1 + (m-1)\beta^2\right)}{\alpha^2 \left(1 + (m-1)\beta^2\right) + (m-1)\beta^2} \\ &\geq \frac{1}{8}. \end{aligned} \tag{25}$$

$$\tag{26}$$

Now since there are at least $\epsilon m$ different such $y$'s, and for each $y$ we can pick a distinct $j$, we get that the sum of the squared norms of the corresponding weights is at least $\epsilon m / 8$. Then we introduce the following lemma.

**Lemma D.2.** *For any $i \neq j$, the following holds for the margin maximizer $\boldsymbol{W}^*$:*

$$\|\boldsymbol{w}_i^*\|^2 + \|\hat{\boldsymbol{w}}_j\|^2 \geq \frac{1}{(1 + \alpha^2)(1 - \beta)^2}$$

*Proof.* Consider any $y, y'$ whose aliases are $(k, c)$ and $(k', c')$ and $y \neq y'$. Firstly, recall the condition in Equation 16 which is

$$\boldsymbol{w}_y^{*\top} \boldsymbol{x}_I - \boldsymbol{w}_{y'}^{*\top} \boldsymbol{x}_I \geq 1, \quad \forall \boldsymbol{x}_I \text{ with label } y.$$

By the condition in Equation 16, we also have

$$\boldsymbol{w}_{y'}^{*\top}\boldsymbol{x}_I' - \boldsymbol{w}_{y}^{*\top}\boldsymbol{x}_I' \geq 1, \ \ \forall \boldsymbol{x}_I' \text{ with label } y'.$$

Note that we assume all examples occur in the training data in the theorem. We let $\boldsymbol{x}_I$ be an example whose $k'$-th element is $\beta$ and $k' + m$-th element is $\beta\alpha$, and let $\boldsymbol{x}_I'$ be an example whose $k$-th element is $\beta$ and $k + m$-th element is $\beta\alpha$, and other elements are the same as in $\boldsymbol{x}_I$. Note that such examples exist. Then we have

$$\|\boldsymbol{x}_I\|^2 = \|\boldsymbol{x}_I'\|^2 = (1+\alpha)^2(1+(m-1))\beta^2)$$
$$-\boldsymbol{x}_I^\top\boldsymbol{x}_I' = -(1+\alpha^2)\beta(2+(m-2)\beta).$$

Similar to the steps from equations 19 to 25, we can solve that

$$\|\boldsymbol{w}_y^*\|^2 + \|\boldsymbol{w}_{y'}^*\|^2 \geq \frac{1}{\|\boldsymbol{x}_I\|^2 + \boldsymbol{x}_I^\top\boldsymbol{x}_I'} = \frac{1}{(1+\alpha^2)(1-\beta)^2}.$$

Note that this hold for any $y \neq y'$ which completes the proof. $\qquad\square$

Combining equation 26 and Lemma D.2 yields $\|\boldsymbol{W}^*\|^2 \geq \frac{\epsilon m}{8} + (1-\epsilon)m\frac{1}{(1+\alpha^2)(1-\beta)^2}$. Then the assumption $\epsilon > \frac{8}{(1+\alpha^2)(1-\beta)^2-8}$ yields $\|\boldsymbol{W}^*\| > \|\hat{\boldsymbol{W}}\|$, which contradicts the fact that $\|\boldsymbol{W}^*\|$ is the solution which separates the data with smallest norm. Therefore $\epsilon \leq \frac{8}{(1+\alpha^2)(1-\beta)^2-8}$. Then we can bound the accuracy on $\mathscr{P}^\star$ by

$$\frac{1}{2} \times 100\% + \frac{1}{2} \times \frac{\epsilon m}{2m} = 50\% + \frac{\epsilon}{4} \leq 50\% + \frac{2}{(1+\alpha^2)(1-\beta)^2 - 8}.$$

### D.2 ANALYSIS FOR MMCL

Here we analyze with the feature masking defined in Definition 5.3. To obtain Theorem 4.8, we just need to set $\pi = 1$.

As shown in Section B.1, the minimizer can be expressed in terms of SVM of $\boldsymbol{S}$. We can calculate that

$$\boldsymbol{S} = C\boldsymbol{D}_I^\top \begin{bmatrix} \frac{1+\pi(m-1)\beta^2}{m}\mathbf{I}_m & \frac{\pi\alpha}{m}\mathbf{I}_m \\ \frac{\alpha}{m}\mathbf{I}_m & \frac{1+(m-1)\beta^2}{m}\pi\alpha^2\mathbf{I}_m \end{bmatrix} \boldsymbol{D}_T, \tag{27}$$

where $C$ is some constant.

Then, at test time, the similarity between an input $\boldsymbol{x}_I = \boldsymbol{D}_I\boldsymbol{z}$ from class $y$ and a prompt $\boldsymbol{p}_{c'} = \boldsymbol{D}_T c' \boldsymbol{e}_{k'}$ from class $(k', y')$ denoted by $Sim$, is given by

$$Sim = C\boldsymbol{z}^\top \left( \frac{1+\pi(m-1)\beta^2}{m}\boldsymbol{e}_{k'} + \frac{\alpha}{m}\boldsymbol{e}_{k'+m} \right).$$

There are two cases to consider. if $(k, c) = (k', c')$

$$Sim_1 = C\left(\frac{1}{m} + \frac{m-1}{m}\pi\beta^2 \pm \frac{\alpha^2}{m}\right). \tag{28}$$

If $k = k', c \neq c'$

$$Sim_2 = -C\left(\frac{1}{m} - \frac{m-1}{m}\pi\beta^2 \pm \frac{\alpha^2}{m}\right) \tag{29}$$

If $k \neq k'$

$$Sim_3 = C\left(\pm\beta\left(\frac{1}{m} + \frac{m-1}{m}\pi\beta^2\right) \pm \beta\frac{\alpha^2}{m}\right) \tag{30}$$

To achieve 100% accuracy, we need $Sim_1 > Sim_2, Sim_1 > Sim_3$. which yields,

$$\pi(m-1) > \frac{\alpha^2-1}{\beta^2} \tag{31}$$

and

$$(1-\beta)\beta^2\pi(m-1) > (1+\beta)\alpha^2 - 1 + \beta. \tag{32}$$

Note that if inequality 32 holds, then inequality 31 holds as well. Therefore we only need inequality 32, which completes Theorem 5.4.

We note that Theorem 4.8 is obtained by setting $\pi = 1$.

# E  ANALYSIS FOR IN-DISTRIBUTION ACCURACY

## E.1  DATA MODEL 1

We provide a brief explanation about the in-distribution accuracy, starting from SL.

W.L.O.G., we can let $\boldsymbol{D}_I$ be the first $l$ columns of the identity matrix. Then the core and spurious features show in the first two elements of the input. Let $\hat{w}_{\text{core}}$ and $\hat{w}_{\text{spu}}$ denote first two elements of the SL loss minimizer $\boldsymbol{W}^* \in \mathbb{R}^{d_I}$, respectively.

In the following, for simplicity, we assume (1) $\|[\hat{w}_{\text{core}}\ \hat{w}_{\text{spu}}]\| = \Omega(1)$; (2) $\hat{w}_{\text{spu}} > 0$. However, it's worth noting that both of these assumptions can be shown to hold with high probability through more involved analysis.

As shown in Proposition 3 in (Sagawa et al., 2020), the following holds

$$\boldsymbol{\Phi}^{-1}(0.006) \geq \frac{1 - (1+c)\gamma^2 - c' - \hat{w}_{\text{spu}} - \hat{w}_{\text{core}}}{\sqrt{\hat{w}_{\text{core}}^2 \sigma_{\text{core}}'^2 + \hat{w}_{\text{spu}}^2 \sigma_{\text{spu}}'^2}} \tag{33}$$

where $c < 1/2000$, $c' < 1/1000$, $\gamma = 9/10$. After some calculation we have $R := \frac{\hat{w}_{\text{spu}}}{\hat{w}_{\text{core}}} > 1.51$. also alignment with noise is bounded.

Lemma 1 in (Sagawa et al., 2020) shows that

$$\|\boldsymbol{W}^*\|^2 \leq u^2 + s^2 \sigma_{\xi,I}'^2 (1+c_1) n_{min} + \frac{s^2 \sigma_{\xi,I}'^2}{n^4} = O(\frac{n}{\sigma_{\xi,I}^2}), \tag{34}$$

where $u = 1.3125$, $s = \frac{2.61}{\sigma_{\xi,I}'^2}$.

Given a test example $(\boldsymbol{x}_I, y)$, where $\boldsymbol{x}_I = \boldsymbol{D}_I \begin{bmatrix} z_{\text{core}} \\ z_{\text{spu}} \end{bmatrix} + \boldsymbol{\xi}_I$, we have

$$|\boldsymbol{W}^* \boldsymbol{x}_I - [\hat{w}_{\text{core}}\ \hat{w}_{\text{spu}}] \begin{bmatrix} z_{\text{core}} \\ z_{\text{spu}} \end{bmatrix}| \leq |\boldsymbol{W}^* \boldsymbol{\xi}_I|. \tag{35}$$

By considering that $\boldsymbol{W}^* \boldsymbol{\xi}_I$ is a Gaussian variable and applying Lemma C.4 , we have

$$|\boldsymbol{W}^* \boldsymbol{\xi}_I| = O(\|\boldsymbol{W}^*\| \sigma_{\xi,I} \sqrt{\frac{\log d_I}{d_I}}), \tag{36}$$

which further gives

$$|\boldsymbol{W}^* \boldsymbol{\xi}_I| = O(n\sqrt{\frac{\log d_I}{d_I}}) \tag{37}$$

by equation 34. Thus with sufficiently large $d$, $|\boldsymbol{W}^* \boldsymbol{\xi}_I| = o(1)$. Given that $\|[\hat{w}_{\text{core}}\ \hat{w}_{\text{spu}}]\|$ is at least constant, the prediction is dominated by $[\hat{w}_{\text{core}}\ \hat{w}_{\text{spu}}] \begin{bmatrix} z_{\text{core}} \\ z_{\text{spu}} \end{bmatrix}$.

Let's begin by considering test examples where $a = y$. By following similar steps as in equations 13 and 14, we can get the accuracy on such examples $\boldsymbol{\Phi}(\frac{1+R}{\sqrt{\sigma_{\text{core}}^2 + R^2 \sigma_{\text{spu}}^2}}) \pm o(1)$. In the scenario where $\sigma_{\text{core}} = 1$ and $\sigma_{\text{spu}} = 0$, given that $R > 1.51$, the above accuracy is $\boldsymbol{\Phi}(2.51) = 99.4\% \pm o(1)$. Given that in the in-distribution test data, such examples occur with probability $p = 1 - o(1)$, the overall in-distribution accuracy is $\boldsymbol{\Phi}(2.51) = 99.4\% \pm o(1)$.

For MMCL, from Section C.2.4, the accuracy on examples where $a = y$ is $1 - \boldsymbol{\Phi}(\frac{-2p - \sigma_{\text{core}}^2}{\sqrt{(1+\sigma_{\text{core}}^2)^2 \sigma_{\text{core}}^2 + (2p-1)^2 \sigma_{\text{spu}}^2}}) \pm o(1)$. Considering that $p_{\text{spu}} = 1 - o(1)$, the in-distribution accuracy is also $1 - \boldsymbol{\Phi}(\frac{-2p - \sigma_{\text{core}}^2}{\sqrt{(1+\sigma_{\text{core}}^2)^2 \sigma_{\text{core}}^2 + (2p-1)^2 \sigma_{\text{spu}}^2}}) \pm o(1)$. In the case where $\sigma_{\text{core}} = 1, \sigma_{\text{spu}} = 0$, the above accuracy is $1 - \boldsymbol{\Phi}(-1.5) \pm o(1) = 93.93\% \pm o(1)$.

Therefore we see that SL has slightly higher in-distribution accuracy.

## E.2 DATA MODEL 2

For data model 2, it is evident that SL can achieve 100% in-distribution accuracy, as under our assumption, every possible example shows in the training set and they are seperable. For MMCL, every example in $\mathscr{P}^{Tr}$ also shows in $\mathscr{P}^\star$. The fact that it achieves 100% accuracy on $\mathscr{P}^\star$, as we have already proved, implies a 100% in-distribution accuracy.

# F  COMPARISON BETWEEN SUPCON, SELF-SUPERVISED-CL AND MMCL

## F.1  A DETAILED DISCUSSION

While the original analysis in the main paper has already thoroughly demonstrated the mechanisms leading to MMCL's robustness, an in-depth comparison between SupCon, Self-Supervised-CL (e.g., SimCLR), and MMCL offers an alternative interpretation of the same findings. We hope this can further illustrate the roles of two crucial elements in MMCL: contrasting between individual examples and multimodality.

**1. SupCon < Self-Supervised-CL: Role of contrasting between individual examples.** Let's compare the representations learned by two different unimodal representation learning techniques: SupCon and Self-Supervised-CL. Although their loss functions are quite similar, Self-Supervised-CL contrasts any two different examples, while SupCon contrasts only those with different labels. We have the following important conclusions: (1) Self-Supervised-CL closely resembles MMCL but within a single modality. Consequently, its learned representations exhibit a similar structure to MMCL's representations in that modality. This includes learning large-variance features and features shared between classes. (2) In contrast, SupCon's representations exacerbate the issue of spurious correlations because it maps both core and spurious features to the same direction in representations, making them entangled. We theoretically demonstrate this in Theorems F.1 and F.2 in the next subsection.

**2. Self-Supervised-CL < MMCL: Role of multi-modality.** Now, one may ask, given that Self-Supervised-CL achieves good representation structures, can it achieve the same level of robustness as MMCL? Well, (1) No, because Self-Supervised-CL solely learns unimodal (e.g., image) representations. To enable classification, we rely on supervised learning on these representations, which, as we have already shown, is not robust. (2) Yes, **only if one could** align representations of language and image modalities for zero-shot classification (bypassing supervised learning) after separately learning representations in each modality with Self-Supervised-CL. However, this essentially falls into the category of MMCL because MMCL precisely performs these two tasks – contrasting individual examples and aligning between modalities –simultaneously. An alternative way to think about this is that even if we adopt MMCL's image representations, training a linear classifier on the representations instead of conducting zero-shot classification would lead reduced robustness. Evidence for this can be found in Figures 14 and 15 in the CLIP paper (Radford 2021), where zero-shot classification outperforms logistic regression/few-shot learning on representations.

## F.2  THEORETICAL ANALYSIS FOR SUPCON

We consider the following supervised contrastive loss, which is naturally analogous to the MMCL loss used in the main paper, and is akin to the linear loss widely adopted in theoretical CL papers in the literature Ji et al. (2021); Nakada et al. (2023).

$$\mathcal{L}_{\text{SupCon}}(\boldsymbol{W}) = -\frac{1}{n}\sum_i \frac{1}{N_i^+}\sum_{j\|i}\left(g(\boldsymbol{x}_{I,i})^\top g(\boldsymbol{x}_{I,j}) - \sum_{k\nmid i}\frac{g(\boldsymbol{x}_{I,i})^\top g(\boldsymbol{x}_{I,k})}{N_i^-}\right) + \frac{\rho}{2}\|\boldsymbol{W}^\top\boldsymbol{W}\|_F^2.$$

where $j \| i$ represents that examples $i$ and $j$ are from the same class, $k \nmid i$ represents that examples $i$ and $k$ are from different classes, $N_i^+ = |\{j \mid j \| i\}|$ and $N_i^- = |\{k \mid k \nmid i\}|$. We consider a linear model $g(\boldsymbol{x}_I) = \boldsymbol{W}\boldsymbol{x}_I$ with $\boldsymbol{W} \in \mathbb{R}^{p\times d}$. The examples are drawn from the training distribution $\mathscr{P}^{Tr}$ within the image modality.

### F.2.1 DATA MODEL 1, SUPCON

Since SupCon solely learns representations, to enable classification, we need to add a classifier $l$ on top of the encoder $g$. we consider a linear classifier $l(\boldsymbol{v}) = \boldsymbol{\beta}^\top \boldsymbol{v}$ where $\boldsymbol{\beta} \in \mathbb{R}^p$. The entire model, consisting of both the encoder and the added classifier, is represented as $l(g(\boldsymbol{x}_I)) = \boldsymbol{\beta}^\top \boldsymbol{W} \boldsymbol{x}$. Its prediction $\hat{y}(\boldsymbol{x}_I)$ is given by $\text{sign}(l(g(\boldsymbol{x}_I)))$. The test accuracy on the true distribution is then denoted as $\text{Acc}_{\mathscr{P}^\star}(\boldsymbol{w}, \boldsymbol{\beta}) := \mathbb{E}_{\boldsymbol{z}, y \in \mathscr{P}^\star, \boldsymbol{x}_I = \boldsymbol{D}_I \boldsymbol{\mu}(\boldsymbol{z}) + \boldsymbol{\xi}_I}[\mathbb{1}(\hat{y}(\boldsymbol{x}_I) = y)]$.

The following theorem demonstrates that SupCon is not robust to distribution shifts, resulting in random chance accuracy across the entire true distribution $\mathscr{P}^\star$ and making almost entirely wrong predictions on examples where $a \neq y$. To simplify the analysis we consider minimizing the population loss in noiseless setting. It's important to note that similar results in general cases can be obtained using concentration bounds similar to those in Sections C.2.2 and C.2.3.

**Theorem F.1.** *Consider Assumption 4.2 and additionally consider the noiseless setting where $\sigma_{\xi,I} = 0$ and let $n \to \infty$. Let $\boldsymbol{W}^*$ be the minimizer of the SupCon loss, then no linear classifier can separate the learned representations of the two classes well. More specifically, $\forall \boldsymbol{\beta}$,*

$$Acc_{\mathscr{P}^\star}(\boldsymbol{W}^*, \boldsymbol{\beta}) \leq 50\% + o(1).$$

*Meanwhile, the model's test accuracy on examples where $a \neq y$ is $o(1)$, i.e., approaching zero.*

*Proof.* First, we define

$$\boldsymbol{S}_{\text{SupCon}} := \frac{1}{2}\Big( \sum_{y \in \{-1,1\}} \bar{\boldsymbol{x}}^{(y)} \bar{\boldsymbol{x}}^{(y)\top} - \sum_{y \in \{-1,1\}} \bar{\boldsymbol{x}}^{(y)} \bar{\boldsymbol{x}}^{(-y)\top} \Big),$$

where $\bar{\boldsymbol{x}}^{(y)} := \frac{1}{n/2} \sum_{\boldsymbol{x}_{I,i} \text{ from class } y} \boldsymbol{x}_{I,i}$ denotes mean of examples from class $y$. Based on our assumption, we can easily calculate $\boldsymbol{S}_{\text{SupCon}}$ as follows

$$\boldsymbol{S}_{\text{SupCon}} = 2\boldsymbol{D}_I \begin{bmatrix} 1 & 2p_{\text{spu}} - 1 \\ 2p_{\text{spu}} - 1 & (2p_{\text{spu}} - 1)^2 \end{bmatrix} \boldsymbol{D}_I^\top.$$

Let $\sum_{i=1}^d \lambda_i \boldsymbol{u}_i \boldsymbol{u} i^\top$ represent the eigen decomposition of $\boldsymbol{S}_{\text{SupCon}}$. Similar to the explanation provided in Section B.1, by rewriting $\mathcal{L}_{\text{SupCon}}$ as a matrix factorization objective and applying the Eckart-Young-Mirsky theorem, we obtain the minimizer of the loss as follows:

$$\boldsymbol{W}^{*\top} \boldsymbol{W}^* = \frac{1}{\rho} \sum_{i=1}^p \lambda_i \boldsymbol{u}_i \boldsymbol{u}_i^\top,$$

We can easily compute the eigen vectors and eigen values for $\boldsymbol{S}_{\text{SupCon}}$: $\lambda_1 = (2p_{\text{spu}} - 1)^2 + 1$, $\boldsymbol{\mu}_1 = \boldsymbol{D}_I \begin{bmatrix} \frac{1}{\sqrt{(2p_{\text{spu}}-1)^2+1}} \\ \frac{2p_{\text{spu}}-1}{\sqrt{(2p_{\text{spu}}-1)^2+1}} \end{bmatrix}$, and $\lambda_2 = \cdots = \lambda_d = 0$. Therefore, $\boldsymbol{W}^* = \sqrt{(2p_{\text{spu}}-1)^2+1}\, \boldsymbol{P}[\frac{1}{\sqrt{(2p_{\text{spu}}-1)^2+1}} \quad \frac{2p_{\text{spu}}-1}{\sqrt{(2p_{\text{spu}}-1)^2+1}}]\boldsymbol{D}_I^\top$ where $\boldsymbol{P}$ can be any $p \times 1$ vector with norm 1. Consequently

$$\boldsymbol{W}^* \boldsymbol{x} = \sqrt{\frac{2((2p_{\text{spu}}-1)^2+1)}{\rho}} \boldsymbol{P}[\frac{1}{\sqrt{(2p_{\text{spu}}-1)^2+1}} \quad \frac{2p_{\text{spu}}-1}{\sqrt{(2p_{\text{spu}}-1)^2+1}}]\boldsymbol{D}_I^\top \boldsymbol{x}$$

$$= \sqrt{\frac{2((2p_{\text{spu}}-1)^2+1)}{\rho}} \boldsymbol{P}[\frac{1}{\sqrt{(2p_{\text{spu}}-1)^2+1}} \quad \frac{2p_{\text{spu}}-1}{\sqrt{(2p_{\text{spu}}-1)^2+1}}]\boldsymbol{z}. \quad (38)$$

For any $\boldsymbol{\beta}$, the test accuracy is given by

$$\text{Acc}_{\mathscr{P}^\star}(\boldsymbol{W}^*, \boldsymbol{\beta}) = \Pr(y\boldsymbol{\beta}^\top \boldsymbol{W}^* \boldsymbol{x} > 0)$$

By equation 38 and some calculation we get

$$\text{Acc}_{\mathscr{P}^\star}(\boldsymbol{W}^*, \boldsymbol{\beta}) = \Pr(\nu_1 + \nu_2 > -1 - (2p_{\text{spu}} - 1)ay)$$

where

$$\nu_1 = y\xi_{\text{core}}$$
$$\nu_2 = y(2p_{\text{spu}} - 1)\xi_{\text{spu}}$$

and each can be considered as a Gaussian random variable with zero mean, independent from each other. Therefore

$$\Pr(\nu_1 + \nu_2 > -1 - (2p_{\text{spu}} - 1)ay) = \frac{1}{2}\Pr(\nu_1 + \nu_2 > -2p_{\text{spu}}) + \frac{1}{2}\Pr(\nu_1 + \nu_2 > 2(p_{\text{spu}} - 1)).$$

Considering $\frac{\nu_1 + \nu_2}{\sqrt{\sigma_{\text{core}}^2 + (2p_{\text{spu}} - 1)^2\sigma_{\text{spu}}^2}} \sim \mathcal{N}(0, 1)$, we have

$$
\begin{aligned}
\text{Acc}_{\mathscr{P}^\star}(\boldsymbol{W}^*, \boldsymbol{\beta}) \leq & \frac{1}{2}\boldsymbol{\Phi}\left(\frac{2p_{\text{spu}}}{\sqrt{\sigma_{\text{core}}^2 + (2p_{\text{spu}} - 1)^2\sigma_{\text{spu}}^2}}\right) + \frac{1}{2}\boldsymbol{\Phi}\left(\frac{2(1 - p_{\text{spu}})}{\sqrt{\sigma_{\text{core}}^2 + (2p_{\text{spu}} - 1)^2\sigma_{\text{spu}}^2}}\right) \\
= & \frac{1}{2}\boldsymbol{\Phi}\left(\frac{2p_{\text{spu}}}{\sqrt{\sigma_{\text{core}}^2 + (2p_{\text{spu}} - 1)^2\sigma_{\text{spu}}^2}}\right) + o(1) \quad \text{because } p_{\text{spu}} = 1 - o(1) \\
\leq & 50\% + o(1).
\end{aligned}
$$

Additionally, the accuracy on examples where $a \neq y$ is given by $\frac{1}{2}\boldsymbol{\Phi}\left(\frac{2(1 - p_{\text{spu}})}{\sqrt{\sigma_{\text{core}}^2 + (2p_{\text{spu}} - 1)^2\sigma_{\text{spu}}^2}}\right) = o(1)$, i.e., almost zero. $\qquad\square$

### F.3 Data Model 2, SupCon

**Theorem F.2.** *Under the same assumption as in Theorems 4.7 and 4.8. Let $\boldsymbol{W}^*$ be the minimizer of the $\mathcal{L}_{SupCon}$ loss. If we train a multi-class linear classifier $g(\boldsymbol{f}) = \boldsymbol{B}\boldsymbol{f}$ with with $\boldsymbol{B}^{2m \times p}$ using Cross-Entropy loss on the learned representations (i.e., given by $\boldsymbol{W}^*\boldsymbol{x}_I$), the test accuracy over the true distribution is 50%. Formally, let $\boldsymbol{B}^*$ be the trained linear classifier $Acc_{\mathscr{P}^*}(\boldsymbol{W}^*, \boldsymbol{B}^*) = 50\%$. Moreover, no linear classifier can achieve accuracy better than 75%, i.e., $\forall \boldsymbol{B}, Acc_{\mathscr{P}^*}(\boldsymbol{W}^*, \boldsymbol{B}) \leq 75\%$.*

*Proof.* First, we define

$$\boldsymbol{S}_{\text{SupCon}} = \frac{1}{2m - 1}\sum_{y=1}^{2m}\bar{\boldsymbol{x}}_I^{(y)}\bar{\boldsymbol{x}}_I^{(y)\top},$$

where $\bar{\boldsymbol{x}}_I^{(y)} := \frac{1}{\text{\# examples from class } y}\sum_{\boldsymbol{x}_I \text{ from class } y} \boldsymbol{x}_I = [c\boldsymbol{e}_k^\top \quad \alpha c\boldsymbol{e}_k^\top]^\top$ for any $y$ with alias $(k, c)$. Here $\boldsymbol{e}_k$ denotes the $k$-th standard basis in $\mathbb{R}^m$. Similar to the previous subsection, by rewriting the objective and applying the Eckart-Young-Mirsky theorem, we obtain

$$\boldsymbol{W}^{*\top}\boldsymbol{W}^* = \frac{1}{\rho}\sum_{i=1}^{p}\lambda_i\boldsymbol{u}_i\boldsymbol{u}_i^\top,$$

where $\frac{1}{\rho}\sum_{i=1}^{d}\lambda_i\boldsymbol{u}_i\boldsymbol{u}_i^\top$ is the eigen decomposition of $\boldsymbol{S}_{\text{SupCon}}$. The eigenvalues and vectors can be calculated as follows:

$$\lambda_i = \frac{2}{2m - 1}(1 + \alpha^2), \quad \boldsymbol{u}_i = \boldsymbol{D}_I[\frac{1}{\sqrt{1 + \alpha^2}}\boldsymbol{e}_i^\top \quad \frac{\alpha}{\sqrt{1 + \alpha^2}}\boldsymbol{e}_i^\top]^\top, \quad \forall i \in [m]$$
$$\lambda_i = 0, \quad \forall i > m.$$

Therefore

$$\boldsymbol{W}^* = \sqrt{\frac{2(1 + \alpha^2)}{2m - 1}}\boldsymbol{P}\begin{bmatrix}\boldsymbol{u}_1^\top \\ \vdots \\ \boldsymbol{u}_m^\top\end{bmatrix},$$

where $\boldsymbol{P} \in \mathbb{R}^{p \times m}$ can be any matrix with orthonormal columns. Consequently, for any example $\boldsymbol{x}_I$ in class $(k, c)$,

$$\boldsymbol{W}^*\boldsymbol{x}_I = \sqrt{\frac{2(1 + \alpha^2)}{2m - 1}}\left(\frac{c}{\sqrt{1 + \alpha^2}} + \text{sign}(\boldsymbol{z}[k + m])\frac{\alpha^2}{\sqrt{1 + \alpha^2}}\right)\boldsymbol{P}\boldsymbol{e}_k. \tag{39}$$

On the training distribution, $\text{sign}(\boldsymbol{z}[k+m])$ is always $c$. Therefore, the linear classifier trained on the representations of the training data would be (recall the relation between CE loss and SVD in Section B.1):

$$\boldsymbol{B}^* = \begin{bmatrix} \sqrt{\frac{2m-1}{2}} \frac{1}{1+\alpha^2} \boldsymbol{e}_1^\top \\ -\sqrt{\frac{2m-1}{2}} \frac{1}{1+\alpha^2} \boldsymbol{e}_1^\top \\ \sqrt{\frac{2m-1}{2}} \frac{1}{1+\alpha^2} \boldsymbol{e}_2^\top \\ -\sqrt{\frac{2m-1}{2}} \frac{1}{1+\alpha^2} \boldsymbol{e}_2^\top \end{bmatrix} \boldsymbol{P}^\top.$$

On the true distribution, $\boldsymbol{B}^*$ can make correct predictions for examples that also show during training. However, for any example $\boldsymbol{x}_I$ from class $y$ with alias $(k, c)$, if its $\boldsymbol{z}$ satisfy that $\boldsymbol{z}[k+m] = -c\alpha$, it can be observed that $(\boldsymbol{B}^*\boldsymbol{W}^*\boldsymbol{x}_I)[y-c] > (\boldsymbol{B}^*\boldsymbol{W}^*\boldsymbol{x}_I)[y]$, leading to incorrect predictions. Therefore, the overall accuracy is 50%.

Next, we analyze the case for arbitrary $\boldsymbol{B}$. For any two classes $(k, -1)$ and $(k, 1)$ in the true distribution, we can group them into four groups denoted by $G_{c, \text{sign}(\boldsymbol{z}[k+m])}$, i.e., based on the combinations of $c$ and $\text{sign}(\boldsymbol{z}[k+m])$. Since $\alpha > 1$, we have

$$\frac{-1}{\sqrt{1+\alpha^2}} - \frac{\alpha^2}{\sqrt{1+\alpha^2}} < \frac{1}{\sqrt{1+\alpha^2}} - \frac{\alpha^2}{\sqrt{1+\alpha^2}} < \frac{-1}{\sqrt{1+\alpha^2}} + \frac{\alpha^2}{\sqrt{1+\alpha^2}} < \frac{1}{\sqrt{1+\alpha^2}} + \frac{\alpha^2}{\sqrt{1+\alpha^2}}.$$

Combining this with equation 39, we conclude that the four groups lie on the same line. Moreover, going from $G_{-1,-}$ to $G_{1,1}$, we pass through $G_{1,-1}$ first and then $G_{-1,1}$. Given this order, no linear model can classify more than three out of the four groups correctly. Since this is true for any pair of classes $(k, -1)$ and $(k, 1)$, it follows that no model can perform better than 75% accuracy on the entire distribution. $\square$

## G  EXPERIMENTAL DETAILS

### G.1  THE SEMI-SYNTHETIC EXPERIMENT

**Image generation.** Firstly, we define three types of blue: dark blue, medium blue, and light blue, as well as three types of red: dark red, medium red, and light red. Then, we modify images in the MNIST dataset. To generate the training data, for each image with a digit between 0 and 4, there is a 99.5% probability that the background will be colored with a random shade of blue from the three types; otherwise, it will be colored with a random shade of red. Similarly, for each image with a digit between 5 and 9, there is a 99.5% probability that the background will be colored with a random shade of red from the three types; otherwise, it will be colored with a random shade of blue. The task is to classify whether the digit in an image falls between 0-4 or 5-9. To generate the test data, the color is uniformly selected from all six colors for each image.

**Caption generation.** We simulate captions using vectors. Each caption is a 200-dimensional vector generated as $\boldsymbol{v} + \boldsymbol{\xi}$, where $\boldsymbol{\xi} \sim \mathcal{N}(0, \frac{1}{2000}\mathbf{I}_{200})$ and $\boldsymbol{v} = [a, b, 0, 0, 0, \ldots, 0]^\top$ with $a, b$ generated as follows.

First, we define a dictionary that assigns a value to each color and digit:

```
DICT = {0: -4.5, 1: -3.5, 2: -2.5, 3: -1.5, 4: -0.5, \\
5: 0.5, 6: 1.5, 7: 2.5, 8: 3.5, 9: 4.5, \\
"dark blue": -2.5, "medium blue": -1.5, "light blue": -0.5,\\
"light red": 0.5, "medium red": 1.5, "dark red": 2.5  }
```

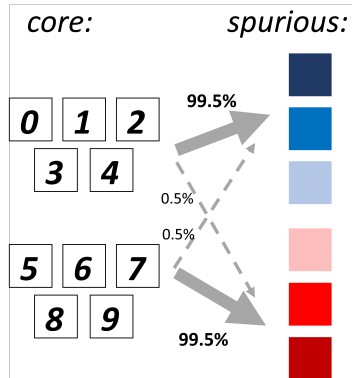

Figure 4: Construction of images

We also define the following values as the means for each category:

$$\text{mean}_{0\text{-}4} = \frac{1}{4}\sum_{d=1}^{4}\text{DICT}[d] = -2.5$$

$$\text{mean}_{5\text{-}9} = \frac{1}{4}\sum_{d=5}^{9}\text{DICT}[d] = 2.5$$

$$\text{mean}_{\text{blue}} = \frac{\text{DICT}[\text{``dark blue"}] + \text{DICT}[\text{``medium blue"}] + \text{DICT}[\text{``light blue"}]}{3} = -1.5$$

$$\text{mean}_{\text{red}} = \frac{\text{DICT}[\text{``dark red"}] + \text{DICT}[\text{``medium red"}] + \text{DICT}[\text{``light red"}]}{3} = 1.5$$

For an image with digit $d$ and color $c$, the corresponding $a$ and $b$ are given by:

$$a = \begin{cases} \text{DICT}[d], & \text{with probability } \pi_{\text{core}} \\ \text{mean}_{0\text{-}4} \text{ if } d \in \{0,1,2,3,4\} \text{ else mean}_{5\text{-}9}, & \text{with probability } 1 - \pi_{\text{core}} \end{cases}$$

$$b = \begin{cases} \text{DICT}[c], & \text{with probability } \pi_{\text{spu}} \\ \text{mean}_{\text{blue}} \text{ if } c \text{ is a kind of blue else mean}_{\text{red}}, & \text{with probability } 1 - \pi_{\text{spu}}. \end{cases}$$

**Training details.** For MMCL, we choose LeNet with an output dimension of 128 as our vision encoder. For the 'language' model, since the captions are represented as vectors, we employ a linear model with an output dimension of 128. We use the CLIP loss, but without normalization when computing similarity. We use momentum SGD as the optimizer with a learning rate of $0.01$, weight decay of $0.001$, momentum of $0.9$, a batch size of 128. The model is trained for 100 epochs. For SL, we train a LeNet using momentum SGD with a learning rate of $0.01$, weight decay of $0.001$, momentum of $0.9$, a batch size of 128, for 40 epoch to minimize the Cross-Entropy loss.

**In distribution accuracy.** To measure the in-distribution test accuracy, we evaluate the models on a dataset constructed in the same way as the training data but with images from the MNIST test set. Figure 5 presents the results, showing that supervised learning achieves the highest in-distribution accuracy. This indicates that the improvement in out-of-distribution accuracy shown in Figure 2 can only be attributed to MMCL's superior robustness. In other words, MMCL enhances both effective robustness and relative robustness, as defined in (Taori et al., 2020).

## G.2 EXPERIMENTS ON MS COCO

**Datasets.** The MSCOCO dataset Lin et al. (2014), or Microsoft Common Objects in COntext, is a comprehensive computer vision dataset comprising diverse images annotated with object masks and captions. We train our model on MSCOCO and test on 6 different variants of ImageNet Russakovsky et al. (2015) as follows. ImageNet1K, a subset of the ImageNet dataset, contains over a million high-resolution images for image classification. ImageNetv2Recht et al. (2019) addresses biases and

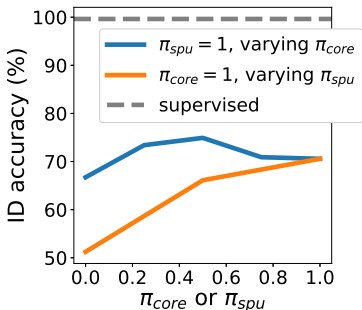

Figure 5: In-distribution test accuracy evaluated on a dataset constructed in the same way as the training data but with images from the MNIST testset.

offers a balanced category distribution. ImageNet Sketch Wang et al. (2019) features hand-drawn object sketches, while ImageNet-A Hendrycks et al. (2021b) serves as an adversarial benchmark. ImageNet-R Hendrycks et al. (2021a) offers real-world, unfiltered images, and ObjectNet Barbu et al. (2019) emphasizes robustness with objects in varied, cluttered scenes.

**Settings.** (1) Training 3-layer MLPs on top of frozen encoders (Figures 3a and 3b). Due to the limited time and compute resource constraints, we consider a simplified setting similar to the one used in Ren & Li (2023). We use the pre-trained ResNet50 of CLIP, followed by a 3-layer fully connected network (MLP) with batch norm between layers for the image encoder part. During training, we freeze the ResNet50 part and solely train the 3-layer MLP. Similarly, for the text part, we employ a separate MLP with the same size on top of the pre-trained Transformer of CLIP and only train the MLP. The hidden and the output dimensions of this MLP are selected from the set $\{16, 32, 64, 128, 256, 512, 768, 1024, 2048, 4096, 8192\}$ to produce different accuracy pairs. We train for 100 epochs with a batch size of 256. Other hyperparameters are set to their default values. Each experiment is run on 1 NVIDIA A6000. (2) Training the entire model (Figure 3c). We use the official Pytorch implementation of CyCLIP Goel et al. (2022)[3]. We train CLIP (with ResNet50 as the image encoder and Transformer as the text encoder) on MSCOCO for 100 epochs with a batch size of 512. Other hyperparameters are set to their default values. Each experiment is run on 2 NVIDIA A6000.

**Evaluation.** Following the measurement proposed in Taori et al. (2020), which has become a standard way of measuring robustness, we plot OOD accuracy (the average of the zero-shot accuracies on 6 shifted versions of ImageNet) against ID accuracy (the zero-shot accuracy on the validation set of MSCOCO). In setting (1), to obtain different ID-OOD accuracy pairs, we train MLPs with 11 different widths as described above. In Setting (2), to obtain different ID-OOD accuracy pairs, we train models on 25%, 50%, 75%, and 100% of the original dataset. Note that, for ImageNet variants, we only sample a subset of classes (320 out of 1000) that can be mapped to the 80 MSCOCO classes.

### G.3 EXPERIMENTS ON CONCEPTUAL CAPTIONS

**Datasets.** Conceptual Caption (CC3M) Sharma et al. (2018) consists of around 3 million image-caption pairs which are collected and processed from Internet webpages. The dataset is divided into Training, Validation, and Test splits. The Training split includes 3,318,333 pairs in which a subset of 2,007,528 has machine-generated labels Ng et al. (2020). Utilizing the image labels in the subset of CC3M's Training split, we filter a subset of CC3M training images whose predicted labels belong to ImageNet classes with a confidence score of at least 0.6. To mitigate the effect of class imbalance, we further select classes with at least 100 but no more than 10,000 samples. The resulting subset consists of 296,801 images corresponding to 316 classes of ImageNet. To create the training and validation datasets, we split the subset with the 7:3 ratio in a stratified fashion.

**Settings.** (1) Training 3-layer MLPs on top of frozen encoders (Figures 3a and 3b). We use the same settings in Appendix G.2 except for the batch size which we increase to 1024 because the training set

---

[3]https://github.com/goel-shashank/CyCLIP

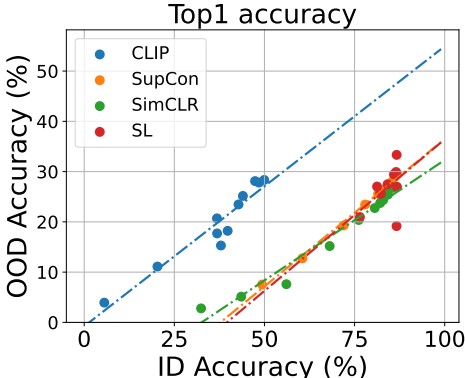

Figure 6: Comparison of robustness (OOD-ID relation) on CC3M between different learning approaches when changing the model width.

is larger. (2) Training the entire model (Figure 3c). We use the same settings as training MSCOCO in Appendix G.2. We cannot increase the batch size in this case due to the GPU memory constraint.

**Evaluation.** We train different models on the training set and perform zero-shot 316-class classification on the validation set for CLIP. Similar to Appendix G.2, all models are evaluated on the test sets of 6 ImageNet variants, and the average accuracy across these datasets is considered the OOD accuracy.

### G.4 ADDITIONAL EXPERIMENTAL RESULTS

**Comparison between different learning algorithms.** In this experiment, we compare CLIP with SL, SupCon, and SimCLR. We train the same MLP as in CLIP on the pre-computed embeddings of images but with different losses for SL, SupCon, and SimCLR. The image embeddings are computed using the pre-trained CLIP image encoder. For a fair comparison, we also train them for 100 epochs with a batch size of 1024. Other parameters are set to the same values as the settings of CLIP. For SupCon and SimCLR, following the standard linear evaluation procedure, we discard the projection head and additionally train a linear classifier on the representations to perform classification. The model width (i.e., the hidden and output dimensions of MLP) varies in the set $\{16, 32, 64, 128, 256, 512, 768, 1024, 2048, 4096, 8192\}$. Figure 6 illustrates that training CLIP yields better robustness than other learning algorithms in terms of zero-shot accuracy. This result is well aligned with our analysis in Appendix F. Interestingly, the robustness of SL, SupCon, and SimCLR is almost similar, resulting in nearly identical slopes.

**Comparison between CLIP and SL with different image encoder's architectures.** In this experiment, we change the architecture of the CLIP image encoder. We utilize four different architectures including ResNet50, ResNet101, ResNet50x4, and ViT-B-32 whose pre-trained weights are available. Figure 7 illustrates that training CLIP on CC3M yields better robustness than training SL across different encoders' architectures.

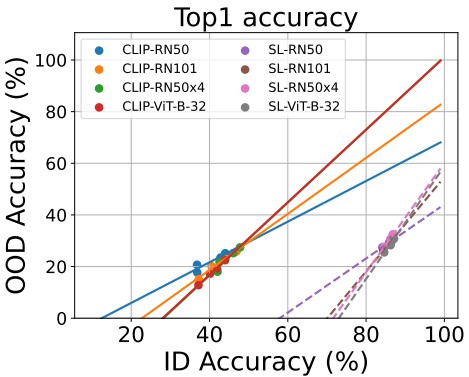

Figure 7: Comparison of robustness (OOD-ID relation) on CC3M between CLIP and SL when changing the image encoder's architecture. The solid lines represent CLIP models while the dashed ones represent SL models.

