# OpenReview forum: "Understanding the Robustness of Multi-modal Contrastive Learning to Distribution Shift"
_ICLR.cc/2024/Conference — ICLR 2024 poster_

### Official Review · Reviewer_RF8u · 2023-10-24

**Soundness:** 3 good
**Presentation:** 2 fair
**Contribution:** 3 good
**Rating:** 5
**Confidence:** 4

**Summary:**

This study delves into the mechanisms behind the success of multimodal contrastive learning (MMCL), particularly in models like CLIP. It identifies two key mechanisms contributing to robustness: intra-class contrasting and inter-class feature sharing. These mechanisms help the model avoid over-reliance on misleading features and enhance its generalization capabilities, particularly in zero-shot classification tasks. The paper also highlights the positive impact of rich, detailed captions for improving model robustness.

**Strengths:**

1. Task importance: This work addresses essential aspects of multimodal contrastive learning, pivotal for AI's real-world performance.

2. Theoretical analysis: The study underscores the necessity of moving beyond surface-level results and engaging in deep theoretical scrutiny to truly understand the mechanisms that drive successful outcomes in contrastive learning.

3. Interesting discoveries: The research brings to light interesting insights, particularly around intra-class contrasting and inter-class feature sharing,

**Weaknesses:**

1. Lack of distinction from traditional approaches: The research does not sufficiently differentiate multimodal contrastive learning (MMCL) from traditional single-modality contrastive learning. It's unclear what unique perspectives MMCL brings compared to these established analyses for conventional contrastive learning. Clarifying this distinction is crucial for understanding the specific contributions and innovations of MMCL.

2. Assumptions about feature learning: The paper posits that intra-class contrasting allows for the learning of high-variance features, leading to the acquisition of generalizable core features beneficial for out-of-distribution (OOD) generalization, especially when annotated with text. However, it does not adequately explain or justify why this effect occurs. A more thorough explanation is needed to understand how intra-class contrasting specifically contributes to OOD generalization.

3. Need for more intuitive theoretical explanations: The theoretical analysis could benefit from more intuitive explanations, examples, or visual aids. As it stands, the theoretical aspects may be challenging for readers to fully grasp, especially those unfamiliar with complex concepts in contrastive learning. Simplifying these explanations could make the research more accessible and understandable.

4. Limitations of experiments: The credibility of the observations is somewhat compromised by the scale and nature of the experiments conducted. The use of small-scale or toy experiments raises questions about the findings' applicability in more complex, real-world scenarios. Future research would benefit from more extensive, large-scale experiments to test the theories in environments that more closely mimic real-world applications.

5. Implications for method design of contrastive learning: The study falls short in outlining how the observations made could influence future designs of contrastive learning methods. While it discusses the mechanisms of MMCL and their benefits, there is a gap in guidance on applying these insights to the practical construction or improvement of next-generation contrastive learning models. Providing clearer, more detailed pathways from observation to application would enhance the paper's utility for future research and development in the field.

**Questions:**

Please address the questions in the above weaknesses.

---

> ### Author Response · Authors · 2023-11-23
>
> We thank the reviewer for acknowledging the importance of the topic and our theoretical contribution, and finding our results interesting.
> > ## “Lack of distinction from traditional approaches ....”
>
> We believe the distinction between MMCL and traditional single-modality CL becomes evident upon comprehending the messages conveyed in our paper. Here, we elaborate further. (1) Any unimodal representation learning, whether supervisedCL (SupCon) or Self-Supervised-CL (like SimCLR), learns an **encoder** rather than a **classifier**. To enable classification, we still need to perform supervised learning by training a classifier on these representations, which, as demonstrated in the paper, is not robust. In contrast, MMCL can bypass supervised learning through zero-shot classification. This, when coupled with the representation structure learned through intra-class contrasting and inter-class feature sharing (analyzed in the Sec 4), leads to robustness. (2) Additionally,we discuss how the richness in captions can contribute to robustness, a factor that doesn’t exist in single-modality CL. (3) **Moreover, we added a new theoretical analysis (Appendix F.2) showing that SupCon maps both core and spurious features into the same direction with similar scales in the representation space**. Consequently, the spurious feature significantly impacts the predictions of a classifier trained on the representations. (3) **We offer a more comprehensive discussion about comparison between MMCL, SupCon, and Self-Supervised-CL in Appendix F.1**. Briefly, the two crucial properties of MMCL—contrasting between individual examples and incorporating multimodality—are both pivotal for robustness, and only MMCL possesses both these aspects. (4) Experimentally, we compare MMCL, SupCon, and Self-Supervised-CL in our new experiments on Conceptual Captions, showing the superior robustness of MMCL. Please refer to the global rebuttal.
>
> > ## Explanation of Intra-class contrasting
>
>
> We believe the mechanism is thoroughly explained and justified in Section 4.1, further validated in experiments shown in figures 2a and 2c. Here, we briefly elaborate to aid understanding of the high-level idea, using the cow-grass example mentioned at the beginning of Sec 4.1. (1) In SL, the large-variance core feature, like a cow's true characteristics, appears unreliable for label prediction due to the variety of cow’s appearances, making it difficult to learn. On the contrary, the low-variance spurious feature, such as grass, is merely a green surface. Hence, the model easily relies on the spurious feature more than the core feature for prediction.(2) In MMCL, the training objective is to contrast unpaired images and texts (e.g., an image of cow A and a text of cow B). With detailed captions that reflect the variance in the images, learning the large-variance core feature aligns with MMCL’s objective of distinguishing different instances of cows. Consequently, the cow feature is no longer considered challenging for the model to learn. For an explanation that delves into more detailed technical aspects, please refer to the text above Thm 4.4. We would be happy to address any specific questions the reviewer may have.
>
>
> > ## Limitations of experiments
>
>
>
> (1) Many other papers already provided the empirical evidence supporting what we're theoretically proving here, on large-scale real-world data. For example, the CLIP paper (Radford 2021, Fig 14) shows the advantage of CLIP over supervised learning; (An 2023) demonstrates that mentioning additional contextual features in captions improves robustness. **Therefore, the focus of our paper is not to replicate observed empirical results, but to provide profound theoretical insights into the underlying mechanisms**. (2) We conducted new experiments on Conceptual Captions, showcasing the benefits of rich captions, and providing a comparison between MMCL and different uni-modal algorithms. These additional experiments, alongside our existing results from MSCOCO, provide further support for our theoretical findings.

---

> ### Author Response · Authors · 2023-11-23
>
> > ## implications of our findings
>
> We emphasize that our paper aims to demystify the robustness of MMCL by unraveling its underlying mechanisms. This study paves the way for further theoretical exploration in this direction. Beyond the theoretical contributions, our research offers valuable practical insights. For instance, given our findings on intra-class contrasting, future research could seek to design loss functions that further enhance intra-class contrasting to achieve even better robustness. Insights into inter-class feature sharing can guide data collection, suggesting that images with multiple objects could contribute significantly to robustness. Moreover, our analysis of rich captions provides guidance on constructing captions for robustness, suggesting that they should mention variance in core features and the shared features among classes. This insight could be used for future research to design better prompts for image (re-)captioning. These implications underscore the broad significance of our work.

---

### Official Review · Reviewer_8jub · 2023-11-01

**Soundness:** 3 good
**Presentation:** 3 good
**Contribution:** 3 good
**Rating:** 6
**Confidence:** 4

**Summary:**

This paper investigates multimodal contrastive learning(MMCL) mechanism that leads to its superior robustness in zero-shot classification/Out-of-distribution tasks. The paper attributes such benefit to MMCL loss that features 1) intra-class contrasting, which enables the model to learn features with high variance (such as core features) than relying on the spurious features that usually have smaller variance (easier to learn as shortcut); 2) inter-class feature sharing such that captions/features in a different class that contains useful information about another class can be learned. Both experiments on synthetic datasets and real-world datasets are performed to validate the theoretical analysis.

**Strengths:**

Overall this paper investigates an interesting problem from a theoretical perspective and provides insights on the mechanism of multimodal contrastive learning and why it learns more robust features compared to supervised learning.

The analytical framework is sound, and the exposition is straightforward.

The conclusions drawn make sense and explain why rich text description is desired and why MMCL tends to be more robust to spurious features compared to supervised learning.

The experiments section includes both synthetic and real-world datasets, and the setup is interesting and validates the theoretical findings.

**Weaknesses:**

For the theoretical analysis part, section 4.6 could benefit from more clarification on the notation and its indications.

For the experiments part, the training dataset (MSCOCO) scale is relatively smaller compared to testing (ImageNet and its variations).

Overall I think this is a sound paper and I have a few questions listed in the Questions section.

Some related works studying multimodal learning and how each modality could interact with and complement each other could be considered to be included in related works such as:

Mind the Gap: Understanding the Modality Gap in Multi-modal Contrastive Representation Learning

Are Multimodal Transformers Robust to Missing Modality?

Understanding and Constructing Latent Modality Structures in Multi-Modal Representation Learning

Investigating why contrastive learning benefits robustness against label noise

**Questions:**

1. In assumption 4.2, is it always true that spurious features have a smaller variance? Sometimes the background features are treated as spurious features and there can be backgrounds that have high variance. How is $\sigma_{spu} = O(1/\sqrt{logn})$ determined?
2. In section 4.2.1, more explanation on the setup and use of notation might help with understanding. Such as what are the indications of $\beta$ and $\alpha$? What is the meaning and usage of c? And what does coordinate k mean?
3. Does rich details in the captions correlate with a high variance of features?
4. In the experiments section, how is the different accuracy on In-domain distribution obtained?
5. As the dataset scale is very different, the accuracy on ImageNet is quite small. Is CLIP trained from scratch on MSCOCO? Usually, CLIP models pre-trained on MSCOCO are evaluated on Flick30K. Are the numbers in Figure 2 reported as the average of ImageNet and its variations? To represent ID and OOD distribution, maybe it makes more sense to pre-train on ImageNet original version and evaluate on domain-shifted versions? In Figure.2b, the accuracy drops a lot as the label approach accuracy increases, does this indicate overfitting on the labels?
6. One comma missing on page 4, the second last line. "independently of each other"

---

> ### Author Response · Authors · 2023-11-23
>
> We thank the reviewer for recognizing our theoretical contributions and the thoroughness of our analysis, as well as finding the experiments interesting and solid.
>
> > ## Further explanation of Definition 4.6
>
> The meanings of k,c,alpha and beta can be found in the paragraph below Definition 4.6. Here we provide a more detailed explanation. (1) Introducing notations k and c serves the convenience of our analysis and statements. Alternatively, we can describe the distribution as follows:  For each label y, if y is odd, then any example in this class consistently has 1 at the (y+1)//2-th coordinate. If y is even, then any example in this class consistently has -1 at the (y+1)//2-th coordinate. $c$ is just the notation we used for the 1 and -1 here and $k$ is just the notation we used for $(y+1)//2$ here for convenience. Hence, the (y+1)//2-th coordinate serves as the core feature for each class. (2) In the training set, each example in class y also holds a spurious feature at the ((y+1)//2+m)-th coordinate, sharing the sign with the core feature but with a larger magnitude $\alpha>1$. Therefore, $\alpha$ represents the strength of the spurious feature. (3) For two different classes $y_1$ and $y_2$, examples in $y_1$ also hold random values at the (y_2+1)//2 -th coordinate, with magnitude $\beta$, and vice versa. Therefore $\beta$ represents the strength of features shared between classes.
>
> Below is a concrete example with four classes when $\beta=0.5$ and $\alpha=2$.
>
> In the true distribution, examples in each class are given by
>
> Class 1: $[1, \pm 0.5, \pm 2, \pm 1 ]$
>
> Class 2: $[-1, \pm 0.5, \pm 2, \pm 1 ]$
>
> Class 3: $[\pm 0.5, 1, \pm 1, \pm 2 ]$
>
> Class 4: $[\pm 0.5, -1, \pm 1, \pm 2 ]$
>
> In the training distribution, examples in each class are given by
>
> Class 1: $[1, \pm 0.5, 2, \pm 1 ]$
>
> Class 2: $[-1, \pm 0.5, -2, \pm 1 ]$
>
> Class 3: $[\pm 0.5, 1, \pm 1, 2 ]$
>
> Class 4: $[\pm 0.5, -1, \pm 1, -2 ]$
>
> > ## The training dataset (MSCOCO) scale is relatively smaller compared to testing?
>
> We note that the training dataset size is larger than the test dataset size. The MSCOCO dataset comprises around 300k images, whereas ImageNet’s validation set (commonly used for testing) consists of only 50k images, not to say that we sampled a subset of classes that can be mapped to MSCOCO classes (320 out of 1000). We also provide other test dataset with different sizes for reference: ImageNetV2 size = 10k, ImageNetR size=30k, ObjectNet size = 50k, ImageNet-Sketch size = 50k, and ImageNet-A size = 7.5k.
>
> > ## New experiments on Conceptual Captions
>
> We added new experiments on Conceptual Captions to enhance our empirical results. Please see the global rebuttal.
>
> > ## Some related works studying multimodal learning and how each modality could interact with and complement each other could be considered to be included in related works such as:
>
> Thanks for providing these related work, we will incorporate them into the related work section in the revised version.
>
>
> > ## Variance of spurious features
>
> (1) To theoretically characterize MMCL’s robustness, we need to consider a setting where spurious features are learned by SL in the first place. Generally, the model tends to rely more on the spurious feature when its variance is smaller than that of the core feature. This has been theoretically analyzed in Sagawa 2020. Regarding backgrounds, although some have larger variance, it's those ‘easier’ backgrounds with smaller variances that are more likely to be learned by the model. (2) The specific value of $\sigma_{spu}$ is a technical assumption, consistent with the setting in Sagawa 2020, enabling us to directly invoke their theorem for supervised learning. However, as mentioned in the footnote on page 5, our MMCL analysis extends to more relaxed assumptions where \sigma_{spu} only needs to be smaller or equal to a constant order, i.e., O(1).

---

> ### Author Response · Authors · 2023-11-23
>
> > ## Does rich details in the captions correlate with a high variance of features?
>
> There are two facets to the richness of captions: mentioning feature variance and mentioning more features, which we have explored in Theorems 5.2 and 5.4, respectively. In Thm 5.2, the large-variance core feature is well-learned only when captions sufficiently mention its variance. If captions fail to delve into the variance of core features and treat all core features of the same kind as a single entity (e.g., use a single word `cow’ to describe all different cows), MMCL cannot attain robustness. In Thm 5.4, we show that mentioning more features generally informs the model about the irrelevance of spurious features. As a result, the model does not learn them, which enhances robustness.
>
>
> > ## In the experiments section, how is the different accuracy on In-domain distribution obtained?
>
> As we mentioned in Appendix E.2, the different OOD-ID accuracy pairs are obtained by varying the subset size. We trained models with 25%, 50%, 75% and 100% of the original dataset.
>
> > ## Experiment on MSCOCO
>
> (1) CLIP is trained from scratch on MSCOCO. (2) As described in Appendix E.2, the numbers in Fig 2 are the average over the six variations of ImageNet. (3) The original ImageNet doesn't allow running CLIP as it has no captions. Recently, Fang et al. (2022) collected captions for ImageNet and conducted experiments comparing CLIP with SL. However, they observed that CLIP achieves an equal level of robustness as SL in this scenario. Therefore, we hypothesize that ImageNet with captions might not be an ideal environment for CLIP to demonstrate its strengths, potentially due to two reasons: a) Images in ImageNet are predominantly simple, without many additional components apart from the main object, limiting inter-class feature sharing, which our research identifies as crucial for robustness; b) Given our findings that richer captions lead to better robustness, the captions available for ImageNet may not be sufficiently rich (e.g., the user descriptions on the Flickr website, from where they collected the captions, are sometimes irrelevant or overly concise). One potential future research direction we believe holds value is to explore generative models to enhance the captions and assess if this leads to improved robustness.
>
>
> > ## In Figure.2b, the accuracy drops a lot as the label approach accuracy increases, does this indicate overfitting on the labels?”
>
>
> In Fig 2b, the legend 'label' refers to the case where we use the image labels as captions. By comparing this with the standard captions, we aim to validate our findings in Theorems 5.2 and 5.4. These theorems suggest that if the captions lack details in the image, such as variance of features and shared features between classes, MMCL cannot achieve robustness. Intuitively, using labels as captions is akin to supervised learning, where intra-class contrasting and inter-class feature sharing, crucial for robustness, are not enabled.
>
>
> > ## Typo on page 4
>
> Thanks for pointing out the typo. We fixed it in the updated PDF.

---

### Official Review · Reviewer_id3b · 2023-11-01

**Soundness:** 3 good
**Presentation:** 3 good
**Contribution:** 3 good
**Rating:** 6
**Confidence:** 2

**Summary:**

This paper is a study on the distribution-shift robustness of multi-modal contrastive learning (MMCL) compared with supervised learning (SL). The authors provide formal definitions of MMCL and SL, and define robustness evaluation metrics for each of them. They provide both theoretical and empirical results in their paper.

**Strengths:**

- The results are shown in both theoretical proofs and empirical experiments, providing a thorough analysis of MMCL and SL.
- For the theoretical parts, the conclusions look well-supported.
- The conclusions are useful in developing new MMCL methods, in the direction of loss function and data filtering and representation.

I am not a machine learning theoretical researcher, and I do not major in this direction. Therefore, I am unable to check the theoretical analysis carefully.

**Weaknesses:**

- There should be more points in Fig.2 to better show the conclusion. Given just 4~5 points, the observation is not very convincing. In Fig.2(b), the linear approximation for "label" is a little far-fetched.
- Only one dataset, namely MSCOCO, is used in "robustness on real data", making the results less convincing. More datasets should be used here.
- For the empirical parts, the authors did not mention which backbone (e.g., ResNet50 or ViT-base) they used for the evaluation. This might significantly affect the resulting accuracy. It is also interesting to investigate the performance difference between different backbones.

**Questions:**

- Is there a way to qualitatively describe the derived or observed conclusions, beyond just "at this robustness is attributed to aspects of the MMCL loss function, i.e. (1) intra-class contrasting (2) inter-class feature sharing, as well as the nature of multi-modal data i.e. (3) richness of captions"?
-The semi-synthetic experiment seems not make much sense, as the task is too simple and the data is limited. Could you please explain the reason for using this dataset?

---

> ### Author Response · Authors · 2023-11-23
>
> We appreciate the reviewer's acknowledgment of the thoroughness of our analysis, the usefulness of our insights, and their recognition that our findings are well-supported.
>
> > ### Additional Experiments for Enhanced Empirical Results
>
> We added new experiments on Conceptual Captions to enhance our empirical results. Please see the global rebuttal.
>
>
> > ### Different architectures
>
> (1) **Architecture for experiments in Fig 2(b)(c).** We used ResNet50 as the image encoder and Transformer as the text encoder. We added this information to Section 6 in the updated PDF (marked in blue).
>
> (2) **Existing empirical results have already shown CLIP's superior robustness across architectures, and that different architectures are on the same line in the OOD-ID plot**. For instance, the original CLIP paper (Radford 2021)'s Fig 13 shows points obtained by different architectures, clearly showing CLIP (purple) outperforming supervised learning (blue) when architecture is varied. Additionally, studies like (Miller 2021, e.g., Fig 5) have indicated that different architectures are on the same OOD-ID line. Hence, given that our comparisons in experiments are based on the OOD-ID relation, we believe these findings hold across architectures.
>
> (3) **We also provide results for different backbone architectures in our new experiments on Conceptual Captions (details in Appendix G)**. Fig 8 shows that the lines for various architectures remain closely aligned under each algorithm, with ViT exhibiting slightly better performance than the others. Overall, CLIP consistently outperforms SL across architectures.
>
>
> > ### “Is there a way to qualitatively describe the derived or observed conclusions …"?
>
> The following key messages are extensively discussed in sections 4 and 5 of the main paper. Here we reiterate them and hope this can enhance comprehension.
>
> (1) Intra-class contrasting: This relates to the design of the loss function. It contrasts each image with captions that are not paired with that image, and vice versa. This strategy enables the learning of large-variance features, because these features are useful for contrasting and minimizing the loss (c.f. Thm 4.4 and the text above it).
>
> (2) Inter-class feature sharing: This relates to both the nature of the data and the loss function. Nuanced details within images from one class, coupled with detailed captions, can offer counterexamples to the spurious correlations found in other classes. MMCL can effectively exploit these counterexamples through its loss function to avoid relying on spurious correlations (c.f. Thm 4.8 and the text above it).
>
> (3) Rich captions: rich captions relate to the nature of textual data and serve as a pivotal element enabling (1) and (2). We've identified two crucial aspects: a) Captions need to mention feature variance to enable (1); and b) Captions should mention more features to facilitate (2). These are discussed in Sec 5.

---

> ### Author Response · Authors · 2023-11-23
>
> > ### Design of the semi-synthetic experiment
>
> The semi-synthetic experiment is carefully designed to conduct controlled experiments for two goals:(1) to compare MMCL and SL’s robustness, and (2) to validate Theorem 4.7, which says that captions mentioning the core feature’s variance can enhance robustness, while also demonstrating the effect of intra-class contrasting.
>
> **This experiment setting mirrors Data Model 1 in Definition 4.1, incorporating feature masking defined in Definition 5.1 to control caption richness**. Here's how it works: The digit is the core feature, while the color is the spurious feature. Image labels are based on whether the digit in the image is <5. Both features exhibit variance: the core feature has five variations within each class (0, 1, 2, 3, 4 in one class, and 5, 6, 7, 8, 9 in the other), while the spurious feature has three variations ('light', 'medium', 'dark') for each color ('blue' and 'red'). In the training set, 99.5% of images in the '<5' class have a random blue background, and the rest have a red background. Similarly, 99.5% of images in the '>=5' class have a random red background, and the remainder have a blue background.
>
> To simplify, we simulate captions using a vector, which can be equivalently considered as follows: Given an image, the caption mentions information about both the core feature (digit) and the spurious feature (color), with the **specificity** controlled by $\pi_{core}$ and $\pi_{spu}$.
>
> - For the core feature, with probability $\pi_{core}$, the caption specifies the exact digit in the image (e.g., 0, 1, 2, etc.). Otherwise, the caption only indicates whether the digit is <5 or >=5: it states '2' for all digits <5 (as 2 represents the mean of 0, 1, 2, 3, 4), and '7' for all digits >=5 (as 7 represents the mean of 5, 6, 7, 8, 9).
>
> - For the spurious feature, with probability $\pi_{spu}$, the caption specifies the exact color in the image (e.g., ‘light blue’, ‘medium red’). Otherwise, it only indicates whether the color is blue or red. We assign each color a number for simplicity.
>
> With this setup, we are able to demonstrate in Fig 2 that MMCL is more robust than SL, and more importantly, that larger $\pi_{core}$ (captions emphasizing the variance in the core feature) is important to achieve such robustness, as predicted by Theorem 5.2.
>
>
> ## Reference
>
> Miller, John P., et al. "Accuracy on the line: on the strong correlation between out-of-distribution and in-distribution generalization." International Conference on Machine Learning. PMLR, 2021.
>
> Radford, Alec, et al. "Learning transferable visual models from natural language supervision." International conference on machine learning. PMLR, 2021.

---

### Official Review · Reviewer_1D1C · 2023-11-03

**Soundness:** 2 fair
**Presentation:** 2 fair
**Contribution:** 2 fair
**Rating:** 5
**Confidence:** 4

**Summary:**

This paper explores the mechanisms behind learning generalizable representations and robustness to distribution shift of existing multi-modal contrastive learning (MMCL) approaches such as CLIP. The theoretical findings attribute the superior robustness to MMCL loss and rich text annotations, specifically to the two mechanisms - (1) intra-class contrasting between image-text pairs, and (2) inter-class feature sharing enabled by MMCL loss. The theoretical findings are validated through experiments using synthetic dataset based on MNIST and robustness study on MS COCo/ImageNet shifted versions.

**Strengths:**

The paper presents a theoretical perspective of the multi-modal contrastive learning (MMCL) method's robustness to distribution shift, supporting the empirical findings from prior works such as CLIP. The work shows insights of using rich text annotations and explains how MMCL loss helps to learn representations with superior robustness against distribution shifts.

**Weaknesses:**

- The paper generalizes training with cross-entropy loss as the supervised learning, but there is class of supervised learning approaches based on contrastive learning/loss [Khosla 2020]. It is not a fair comparison to say the study is about supervised learning and MMCL
- The study is based on a linearized contrastive loss function, so it not clear what is the effect of other variations of contrastive loss functions. Also, there is no supporting validation on if the robustness is due to single modality v/s multi-modality or cross-entropy v/s contrastive loss.
- Lack of details on the experimental study. I suggest authors provide details on experimental setup and distribution shift study/datasets (at least in the appendix, if not able to provide all details in main manuscript)

[Khosla 2020] Khosla, Prannay, et al. "Supervised contrastive learning." Advances in neural information processing systems 33 (2020): 18661-18673.

**Questions:**

- What are the six versions of shifted ImageNet used for evaluation? Is the entire test set from ImageNet-A, ImagetNet-Sketch, ImageNet-v2, ImageNet-R and ObjectNet are used for evaluation?
- Why not consider domain generalization benchmarks such as domainbed [Gulrajani 2020] for distribution shift robustness evaluation?
- I encourage authors to consider including prior works in the contrastive learning paradigm in the related works section for the readers to get the broader perspective on contrastive learning.

[Gulrajani 2020] Gulrajani, Ishaan, and David Lopez-Paz. "In Search of Lost Domain Generalization." International Conference on Learning Representations. 2020.

---

> ### Author Response · Authors · 2023-11-23
> **Response to Reviewer 1D1C -- Part 1**
>
> We thank the reviewer for acknowledging our theoretical contribution and the usefulness of our insights. Below are the answers to their questions.
>
>
> > ### What about SupCon?
>
> (1) **CE loss is inevitably used even for SupCon**. We note that, our paper focuses on the classification task, in which context supervised learning typically means the standard definition that refers to using CE loss to train a **classifier**. It's important to note that SupCon is a supervised representation learning strategy, generating an **encoder** rather than a **classifier**. To enable classification, one still needs standard supervised learning, i.e., **using the CE loss to train a classifier on the representations**. Consequently, the CE loss analyzed in our paper inevitably becomes a part of this process. In a broader context, the CE loss is essentially used after any unimodal representation learning to enable classification. In this sense, our analysis covers a wide range of scenarios, including both classical supervised learning with CE loss, and any unimodal representation learning followed by supervised learning with CE loss. Moreover, the majority of the experiments in the original CLIP paper (Radford 2021) focus on the advantage of CLIP over standard supervised learning with CE loss (which is what we study in our paper).
>
> (2) **Our paper provides ample evidence and reasoning that can be used to deduce why SupCon lacks robustness, making it quite evident.** Referring to Section 4.1, SupCon doesn't perform intra-class contrasting; it brings all representations within the same class closer together. Referring to Section 4.2, SupCon doesn't utilize inter-class feature sharing as it solely relies on labels, thus ignoring additional information, such as the example we discussed regarding 'non-white trees in images of wolves'. The shortcomings of SupCon are also evident in Section 5, discussing the role of richer captions. Theorem 5.3 suggests that with labels as captions ($\pi$ = 0), the robustness significantly decreases – a setup closely resembling SupCon. This is empirically confirmed in Figure 2b.
>
> (3) **To be more precise, in Appendix F.2 in the updated PDF, we offer a new in-depth theoretical analysis for SupCon with both data models 1 and 2, demonstrating its lack of robustness similar to CE.** The analysis shows how SupCon maps both core and spurious features to the same direction in representations, making them entangled. Hence, the spurious feature significantly impacts the predictions of a classifier trained on the representations.
>
>
> (4) **We also compare MMCL, SupCon, SimCLR, and SL in our new experiments on Conceptual Captions**. The results demonstrate that MMCL exhibits better robustness compared to the uni-modal algorithms. Please refer to the global rebuttal for details.
>
> (5) We offer a more involved and comprehensive discussion about comparison between **MMCL, SupCon, and Self-Supervised-CL** in Appendix F.1, to provide a deeper understanding and an alternative perspective on the messages delivered in the main paper.
>
>
> > ### the use of linearized loss in theoretical analysis
>
> (1) **Theoretically**:  As mentioned at the bottom of page 3, the linearized loss and its variations, sharing similar essence (where log exp is replaced by the linear function y=x), are widely utilized in theoretical studies and effectively capture key aspects of various self-supervised losses, including self-supervised CL (Ji 2021, Haochen 2021, Haochen 2022, Shen 2022), MMCL (Nakada 2023), non-contrastive (Liu 2021), and Supervised CL (Xue 2023). Theoretical works by (Tian 2022, Nakada 2023) have proven that when minimizing any contrastive loss using gradient descent, at each step, the direction of GD is the same as minimizing the linearized loss but with reweighted terms, illustrating the pivotal role of the linearized loss. (Garrido 2023) demonstrated through large-scale experiments that many design choices of popular loss functions, including log exp and cos similarity, are unnecessary. All of these suggest that the linearized loss captures the fundamental essence of CL or self-supervised learning in general.
>
> (2) **Empirically**: Our experiments on real data with standard CLIP loss demonstrate that our theoretical findings indeed hold for practical losses. Moreover, many other papers already provided the empirical evidence supporting what we're theoretically proving here, on large data and non-linear loss. For example, the CLIP paper (Radford 2021, Fig 14) shows the advantage of CLIP over supervised learning; (An 2023) demonstrates that mentioning additional contextual features in captions improves robustness.
>
> Therefore, considering both theoretical and empirical perspectives, we expect that the same results hold for non-linear losses with a more complex analysis.

---

> ### Author Response · Authors · 2023-11-23
> **Response to Reviewer 1D1C -- Part 2**
>
> > ### whether the robustness is due to single modality v/s multi-modality or cross-entropy v/s contrastive loss.
>
> This question has actually been comprehensively answered in the paper with strong support for the answer. The conclusion is: both **multi-modality** and  **intra-class contrasting (different than the type of contrasting in SupCon)** are important in achieving robustness. The paper delves into their complex interactions in Sec 4 and 5. Here, we provide a simplified explanation by considering two scenarios: (1) **multimodality w/o intra-class contrasting**: this is exactly the setting considered in Thm 5.3 and the experiment in Figure 2b, where we only use labels as captions, meaning that no intra-class contrasting. We have already shown that the model is not robust in this case. (2) **intra-class contrasting w/o multimodality (e.g., without raw-text captions)**: In classification tasks, this essentially reduces to the standard supervised learning (typically using CE loss) analyzed in the paper. This is because: while algorithms like self-supervised CL do involve contrasting, they only learn an encoder, instead of a classifier. Therefore, to use these representations, we eventually need to train a classifier with CE on the representations, which will still spuriously associate features in the representations with labels. In contrast, multimodality enables zero-shot classification, bypassing the need for learning a classifier with supervised CE loss, as we discussed at the end of Sec 3.2.
>
>
> > ### Details on the experiments
>
> The details, including information about the distribution shift datasets used, were provided in Appendix E.1 and E.2. In the MSCOCO experiment, the six shifted datasets are widely utilized in other studies on distribution shift, such as Taori 2020 and Fang 2022. Should the reviewer have any further specific questions regarding the experimental details, we would be happy to answer them.
>
>
> > ### Why not consider domain generalization benchmarks such as domainbed [Gulrajani 2020] for distribution shift robustness evaluation?
>
> The dataset referenced by the reviewer cannot be used to run MMCL as the training datasets in Domainbed don’t have captions. Instead we consider MSCOCO, widely used in multimodal learning studies (e.g., Santurkar 2022, Goel 2022, Ren 2023, Bansal 2023), and use shifted versions of Imagenet as evaluation set to include a wide range of natural distribution shifts, following previous works such as the original CLIP paper, Taori 2022, and Fang 2022.
>
> > ### I encourage authors to consider including prior works in the contrastive learning paradigm in the related works section for the readers to get the broader perspective on contrastive learning.
>
> Thanks for the suggestions. We added a paragraph in Sec 2 highlighted in blue to briefly (due to the limited space) discuss more related works.
>
> ## Reference
>
> Garrido, Quentin, et al. "On the duality between contrastive and non-contrastive self-supervised learning." ICLR 2023.
>
> HaoChen, Jeff Z., et al. "Provable guarantees for self-supervised deep learning with spectral contrastive loss." Advances in Neural Information Processing Systems 34 (2021): 5000-5011.
>
> Liu, Hong, et al. "Self-supervised learning is more robust to dataset imbalance." arXiv preprint arXiv:2110.05025 (2021).
>
> HaoChen, Jeff Z., and Tengyu Ma. "A theoretical study of inductive biases in contrastive learning." arXiv preprint arXiv:2211.14699 (2022).
>
> HaoChen, Jeff Z., et al. "Beyond separability: Analyzing the linear transferability of contrastive representations to related subpopulations." Advances in Neural Information Processing Systems 35 (2022): 26889-26902.
>
> Shen, Kendrick, et al. "Connect, not collapse: Explaining contrastive learning for unsupervised domain adaptation." International Conference on Machine Learning. PMLR, 2022.
>
> Xue, Yihao, et al. "Which Features are Learnt by Contrastive Learning? On the Role of Simplicity Bias in Class Collapse and Feature Suppression." arXiv preprint arXiv:2305.16536 (2023).
>
> Tian, Yuandong. "Understanding deep contrastive learning via coordinate-wise optimization." Advances in Neural Information Processing Systems 35 (2022): 19511-19522.
>
> Santurkar, Shibani, et al. "Is a caption worth a thousand images? a controlled study for representation learning." arXiv preprint arXiv:2207.07635 (2022).
>
> Ren, Yunwei, and Yuanzhi Li. "On the Importance of Contrastive Loss in Multimodal Learning." arXiv preprint arXiv:2304.03717 (2023).
>
> Goel, Shashank, et al. "Cyclip: Cyclic contrastive language-image pretraining." Advances in Neural Information Processing Systems 35 (2022): 6704-6719.
>
> Bansal, Hritik, et al. "CleanCLIP: Mitigating Data Poisoning Attacks in Multimodal Contrastive Learning." arXiv preprint arXiv:2303.03323 (2023).

---

### Author Response · Authors · 2023-11-23

We thank the reviewers for recognizing our contribution in unraveling the mechanisms behind significant topics, appreciating the depth of our analysis, acknowledging the utility of our insights, and finding that our conclusions are well-supported by empirical results.

We added a new experiment on a subset of Conceptual Captions (CC) to provide further empirical results for our findings. Specifically, we trained models on Conceptual Caption and evaluated on shifted versions of ImageNet, and compared the OOD-ID plot for different algorithms. Further details are provided in Appendix G. The results are summarized as follows

(1) **Comparison between CLIP, SupCon, SimCLR and SL**. In Fig 6, the OOD-vs-ID line for CLIP surpasses all others. Additionally, the lines for uni-modal CL, including supervised (SupCon) and self-supervised (SimCLR), are very close to that of SL. This further supports our analysis and discussion in the rebuttal and Appendix F, highlighting the shortcomings of uni-modal supervised/self-supervised CL due to its inevitable use of CE loss for classification.

(2) **Benefits of rich captions**. We replicated our experiment from MSCOCO (Fig 2(b)) to CC. Figure 7 presents the comparison between training CLIP with original captions (labeled as ‘standard caption’) and training CLIP with labels as captions (labeled as ‘label as caption’). We make the same observation as before, that ‘standard caption’ achieves better robustness, which once again confirms the conclusions drawn from our theoretical analysis in Sec 5, emphasizing the advantage of richer captions for enhancing robustness. Notably, other studies, such as the large-scale experiments in (Nguyen 2023), also offer empirical support for our theoretical findings.

---

### Meta-Review · Area_Chair_5sqi · 2023-12-18

**Metareview:**

The paper provides two mechanisms to explain the robustness of multimodal models like CLIP, going beyond the usual explanation of "more data". The two mechanisms are clearly described in both prose and the theoretical model. The experimental results also attempt to verify these mechanisms in practice. The reviewers asked for more experiments which the authors provided in the rebuttal. I rate this paper as borderline accept because I think the insights are neat, and offer some new perspectives on why CLIP models are more robust. I also agree with the reviewer concerns regarding the need for more illustrations of the effects rather than just theoretical statements, and some reflections on how this could inspire new methods.

**Justification For Why Not Higher Score:**

Concerns regarding clarity of presentation, and empirical grounding and significance in informing future research around robustness from multimodality

**Justification For Why Not Lower Score:**

I think the claims are insightful and offer a fresh and more complete perspective (compared to prior work) on what makes CLIP and other such models robust

---

### Decision · Program_Chairs · 2024-01-16

Accept (poster)